# Internal Causal Mechanisms Robustly Predict Language Model Out-of-Distribution Behaviors

**Jing Huang** [* 1]  **Junyi Tao** [* 1]  **Thomas Icard** [1]  **Diyi Yang** [1]  **Christopher Potts** [1]

## Abstract

Interpretability research now offers a variety of techniques for identifying abstract internal mechanisms in neural networks. Can such techniques be used to predict how models will behave on out-of-distribution examples? In this work, we provide a positive answer to this question. Through a diverse set of language modeling tasks—including symbol manipulation, knowledge retrieval, and instruction following—we show that the most robust features for correctness prediction are those that play a distinctive causal role in the model's behavior. Specifically, we propose two methods that leverage causal mechanisms to predict the correctness of model outputs: *counterfactual simulation* (checking whether key causal variables are realized) and *value probing* (using the values of those variables to make predictions). Both achieve high AUC-ROC in distribution and outperform methods that rely on causal-agnostic features in out-of-distribution settings, where predicting model behaviors is more crucial. Our work thus highlights a novel and significant application for internal causal analysis of language models.

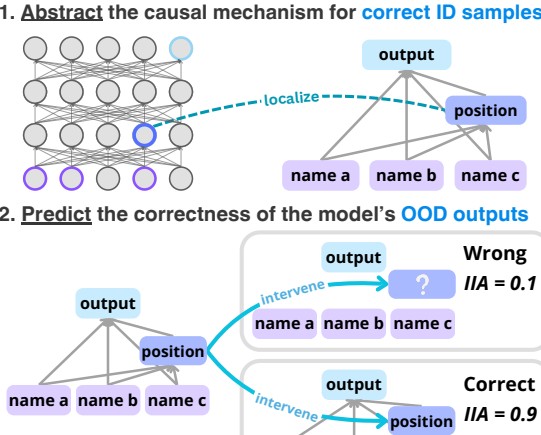

*Figure 1.* We propose a two-stage framework to predict correctness of model outputs under distribution shifts, illustrated using the counterfactual simulation method on the Indirect Object Identification task (Wang et al., 2023). **Stage 1 (Abstract)**: Identify the abstract causal mechanisms the model uses to solve the task correctly on in-distribution data. **Stage 2 (Predict)**: Predict the model's output correctness on out-of-distribution inputs by checking whether it implements the same mechanisms. **Dark blue** indicates the key causal variables used for correctness prediction.

## 1 Introduction

A core task in interpretability research is to identify internal mechanisms that mediate a model's input–output behavior. The motivations for this focus are several. One is simply that compressed, algorithmic descriptions of complex models can help confer a sense of understanding insofar as the algorithms themselves are relatively simple and intuitive. Another, related motivation stems from an intuition that the very presence of abstract algorithmic mechanisms is deeply related to a model's ability to *generalize* beyond the specific domains in which it has been trained. Indeed, it is commonly

---

[*]Equal contribution  [1]Stanford University. Correspondence to: Jing Huang <hij@stanford.edu>.

*Proceedings of the 42ⁿᵈ International Conference on Machine Learning*, Vancouver, Canada. PMLR 267, 2025. Copyright 2025 by the author(s).

assumed that a model will succeed at a generalization task if and only if it has induced a mechanism that implements a "correct" algorithm for that task (provided one exists).

There are two directions to this hypothesized correspondence. In one direction, when a model is successful, this warrants a search for the internal mechanisms that explain success. That is, we search for an appropriate *abstraction* of the model to elucidate how the model was able to generalize (Geiger et al., 2024a). This is the focus of interpretability work that aims to understand the internal causal mechanisms that mediate LLM behaviors (Vig et al., 2020; Geiger et al., 2021; Finlayson et al., 2021; Olsson et al., 2022; Wang et al., 2023; Meng et al., 2023; Wu et al., 2023).

It would be desirable if we were able to move in the other direction as well, from identifying an abstract mechanism to better predicting model generalization behaviors. We refer to this direction as *prediction*. In this work, we explore a

specific instance of the prediction direction—predicting the correctness of LLM behaviors under distribution shifts.

As language models are increasingly deployed in high-stakes, open-ended scenarios, it is critical that we have a handle on how to prevent incorrect or unsafe outputs across a wide range of unanticipated user inputs. However, behavioral testing usually cannot be comprehensive given the compositionally large input space of a natural language task and the potential for distribution shifts across deployed environments. The traditional approach is to estimate the correctness of model predictions using confidence scores (Hendrycks & Gimpel, 2017; Lakshminarayanan et al., 2017; Guo et al., 2017). A more recent line of work uses internal representations of the model (usually supervised probes or use other statistical measures of model activations) to predict the model's correctness on factual retrieval tasks (Azaria & Mitchell, 2023; Zou et al., 2023; Orgad et al., 2025; Marks & Tegmark, 2024; Gottesman & Geva, 2024; Ferrando et al., 2025; Ji et al., 2024; Yin et al., 2024; Chuang et al., 2024).

With the present paper, we seek to show that causal interpretability methods can play a critical role in correctness prediction. Our central question is the following: if we know what internal features causally contribute to the model's predictions in the in-distribution setting, can we use them as robust predictors of the model's out-of-distribution behaviors? To answer this question, we propose two methods that leverage previously identified causal mechanisms to predict model generalization behavior (as illustrated in Figure 1). The first method is *counterfactual simulation*, which predicts whether the model output is correct by checking whether it computes the key causal variables needed for solving the task. The second method is *value probing*, which predicts correctness based on the decision boundaries of different values taken by a causal variable.

We compare our proposed methods against existing methods across a diverse set of five tasks under in-distribution (ID) and multiple out-of-distribution (OOD) settings. The baseline methods include traditional confidence score-based methods and causal-agnostic probing methods using internal activations. The evaluated tasks range from highly curated symbolic tasks to more open-ended tasks where only partial mechanisms have been identified, including Indirect Object Identification (Wang et al., 2023), PriceTag (Wu et al., 2023), RAVEL (Huang et al., 2024), MMLU (Hendrycks et al., 2021), and UnlearnHP (Thaker et al., 2024). Our results show that methods leveraging task-relevant causal features achieve high AUC-ROC scores under ID settings—comparable to or slightly better than the confidence score baseline. Crucially, these causal feature-based methods significantly outperform existing baselines in OOD settings. Our strongest method, counterfactual simulation, improves

average AUC-ROC by 13.84% over prior baselines.

Overall, our results show that internal causal features are more robust predictors of correctness than non-causal ones, especially under distribution shifts. More broadly, our findings highlight a novel and important application of causal interpretability analysis, suggesting a path from understanding model internals to improving model safety and reliability.

## 2 Background

### 2.1 Finding Abstract Causal Mechanisms in LLMs

A trained neural network can be construed as a causal model, with variables being the neurons in the network and the functional mechanisms being given by weight matrix multiplications across the layers that comprise the neural network. Causal interpretability research asks the question: Is there a *more abstract* causal model that adequately captures the structure of the neural network when it comes to a particular task that the network successfully performs (Geiger et al., 2021; 2024a)? This line of work is built on *causal abstraction*, which is concerned with the general question of when one causal model can be said to *abstract* another (Rubenstein et al., 2017). In the interpretability setting, the least abstract causal model is just the neural network itself, and the more abstract causal model is referred to as *high-level* model. Below we review the workflow of finding such abstractions using the language of causal abstraction.

**Proposing high-level causal model** High-level causal models should capture the shared structure among examples of the task, such as the underlying algorithmic description of the task solution. Such models are typically proposed by researchers based on prior knowledge of the task. Some recent work attempts to reduce reliance on human priors by automating this process (Hernandez et al., 2022; Bills et al., 2023; Sun et al., 2025). In this work, we adopt high-level models that have been proposed and verified in the literature.

**Localizing high-level variables in neural networks** Given a high-level model, we search for its implementation in the neural network. Existing methods localize each high-level variable to the model representation space through a bijective function (Huang et al., 2024; Geiger et al., 2024a).

We focus on one such method, Distributed Alignment Search (DAS; Geiger et al. 2024b), which achieves the highest accuracy on localization benchmarks (Huang et al., 2024; Arora et al., 2024; Mueller et al., 2025). DAS uses *distributed interchange interventions* to identify an orthonormal basis $Q$ as the bijective function. Let $\texttt{GetVal}(\mathcal{M}(x), R)$ be the value of some model representation $R$ when the model $\mathcal{M}$ processes an input $x$, and let $\mathcal{M}_{R \leftarrow r}(x)$ be the output of $\mathcal{M}$ when it processes $x$, but

with $R$ set to value $r$. Given a pair of inputs $x_{base,i}$ and $x_{src,i}$, we perform an intervention on the subspace that localizes the variable:

$$r_{base,i} = \texttt{GetVal}(\mathcal{M}(x_{base,i}), R)$$
$$r_{src,i} = \texttt{GetVal}(\mathcal{M}(x_{src,i}), R)$$
$$r_{inv,i} = (I - Q^\top Q)\, r_{base,i} + Q^\top Q\, r_{src,i} \qquad \text{(II)}$$

To find the optimal $Q$, DAS minimizes loss $\ell_Q$ over $N$ counterfactual pairs $D_{cf} = \{(x_{base,i}, x_{src,i}, y_{cf,i})\}_{i=1}^{N}$, i.e., a pair of randomly sampled inputs and a counterfactual label determined by the high-level model, as shown in Eq II.

**Distributed Alignment Search**

$$\ell_Q = \mathbb{E}_{1 \le i \le N}\big[ -y_{cf,i} \cdot \log y_{inv,i} \big]$$
$$\text{where} \quad y_{inv,i} = \mathcal{M}_{R \leftarrow r_{inv,i}}(x_{base,i}) \qquad \text{(DAS)}$$

We define the metric *interchange intervention accuracy* (IIA) to measure the extent to which an implementation exists in $\mathcal{M}$:

$$\texttt{IIA}(D_{cf}) = \mathbb{E}_{1 \le i \le N}\big[\tau(y_{cf,i}) = \tau(y_{inv,i})\big] \qquad \text{(IIA)}$$

where $\tau$ maps the low-level representation to a high-level value, e.g., decoding logits into a token.

## 2.2 Correctness Estimation

Correctness estimation is a fundamental problem in machine learning evaluation and reliability assessment. It is the problem of predicting whether a model output matches the ground truth. For a classifier, a natural choice is to use probabilistic predictions, i.e., the confidence scores (Hendrycks & Gimpel, 2017; Lakshminarayanan et al., 2017). However, LMs' outputs are structured predictions which add complexities to calibration (Kuleshov & Liang, 2015), and deep neural models are only well-calibrated to in-domain inputs after temperature scaling (Guo et al., 2017; Desai & Durrett, 2020; Kadavath et al., 2022; OpenAI, 2024; Wang et al., 2024). Alternatives have been proposed, such as using verbalized confidence scores for instruction-tuned models (Kadavath et al., 2022; Tian et al., 2023). Most related to ours is the line of work using internal representations to estimate the knowledge stored in LMs (Gottesman & Geva, 2024; Marks & Tegmark, 2024; Ferrando et al., 2025), assess the risk of hallucination (Azaria & Mitchell, 2023; Zou et al., 2023; Du et al., 2024; Chuang et al., 2024; Ji et al., 2024; Yin et al., 2024; Orgad et al., 2025), and predict jailbreaking behaviors (Arditi et al., 2024). However, identifying robust features from internal representations that generalize across tasks and datasets remains a challenge (Orgad et al., 2025). This is the problem that we focus on in this work.

## 3 Problem Setting

We are interested in predicting whether a language model $\mathcal{M}$ produces correct answers on a task under distribution shifts. We formalize this as a correctness prediction problem.

**Task** We first define our task as a function that takes as the input a sequence of variables $\mathcal{X}_1, \ldots, \mathcal{X}_n$ and returns an output $\mathcal{Y}$. For symbolic tasks, the function is the symbolic algorithm; for knowledge retrieval tasks, the function is simply a look-up table, mapping each input to an output. Next, we define a prompt template $T$ and use it to generate all valid prompts by assigning values to the input variables. We focus on tasks whose correct outputs are completely determined by processing the input variables using the defined task function. Everything else in the template, besides the input variables, serves merely as background conditions.

Take the PriceTag task as an example (illustrated in Figure 2). The input variables are $\mathcal{X}_{lb}$, $\mathcal{X}_{ub}$, and $\mathcal{X}_q$, representing the lower bound, the upper bound, and the query. The task function is given by $f(\mathcal{X}_{ub}, \mathcal{X}_{lb}, \mathcal{X}_q) = [\![\mathcal{X}_{ub} > \mathcal{X}_q]\!] \wedge [\![\mathcal{X}_{lb} < \mathcal{X}_q]\!]$. We use the template $T =$ "Does the following item cost between $\${\texttt{lower\_bound}}$ and $\${\texttt{upper\_bound}}$? Item: $\${\texttt{query}}$." to generate all prompts, where each input variable is assigned a number sampled from $[0, 20]$.

**ID vs. OOD setting** For each task, we construct one in-distribution and multiple OOD settings. For the in-distribution setting, we fill the input variables and the template with a set of values frequent in pre-training corpora. For the OOD settings, we either introduce perturbations to background conditions irrelevant to solving the task (e.g., replacing the dollar sign with a lira sign in PriceTag) or change the values of input variables (e.g., changing the language of names in IOI). These changes empirically lead to a decline in task accuracy compared with the in-distribution setting. The assumption is that if the task function remains the same, a model employing a systematic solution should continue to make correct predictions despite irrelevant changes.

**Problem formulation** Given a task input $x_i$, we compare the model's prediction $\mathcal{M}(x_i)$ to the ground-truth label, which yields a correctness label $l_i \in \{0, 1\}$, where 1 indicates correct. The resulting dataset $D = \{(x_i, \mathcal{M}(x_i), l_i)\}_{i=1}^{N}$ consists of $N$ tuples of inputs, model predictions, and correctness labels. As most prediction methods are supervised, we additionally assume access to a labeled subset of $M$ ID examples which we call the *verified set*: $\mathbb{V} = \{(x_i, \mathcal{M}(x_i), l_i)\}_{i=1}^{M}$. At test time, we evaluate our methods on both in-distribution and OOD settings. To do this, we sample unlabeled test inputs $x_{test}$ from the ID dataset (excluding the verified set) and the OOD datasets. All samples are drawn independently and identically. The correctness prediction problem is thus formulated as a binary classification task predicting $P(l = 1 \mid \mathcal{M}, x_{test})$.

# 4 Methods

In this section, we first review existing methods and then introduce two new methods that leverage internal causal mechanisms to predict correctness. We categorize these methods into three families based on the types of features they use: (1) output probabilities, (2) *internal causal-agnostic features* that are not informed by causal understandings of the model's internal mechanisms, and (3) *internal causal features* that correspond to identified abstract causal variables.

## 4.1 Output Probabilities

The first family of methods uses output probabilities as features for predicting output correctness, essentially treating them as confidence scores. The predictive power of confidence scores is grounded in the modeling objective—in a perfectly calibrated model, the predicted probability is exactly equal to the expected accuracy. Since LMs are not always perfectly calibrated, we additionally apply temperature scaling to confidence scores for a fairer evaluation of their predictiveness (OpenAI, 2024; Kadavath et al., 2022).

Existing work typically aggregates token-level confidence scores to predict the correctness of an output sequence. Given a test input $x_{test}$, let $t_1, \ldots, t_n$ be the sequence of output tokens from $\mathcal{M}$. Denote the confidence score $c_i$ as the probability of token $t_i$ at decoding step $i$, computed as logits divided by the temperature parameter $T$. The correctness $f(\mathcal{M}, x_{test})$ is approximated as the log-likelihood of a subset of output tokens $S$, normalized by its size.

**Confidence Scores**

$$f(\mathcal{M}, x_{test}) = \frac{1}{|S|} \log \left( \prod_{c_i \in S} c_i \right) = \frac{1}{|S|} \sum_{c_i \in S} \log(c_i) \quad (1)$$

There are two common variations that differ by how they select the set of tokens in $S$. The simplest one, **First n Tokens**, uses the first $n$ decoded tokens, where $n$ is a hyperparameter. An alternative is to select tokens that represent the actual answer, which we refer to as **Answer Tokens**.

Since AUC-ROC evaluation is ranking-based, we can directly use the log-likelihood without further normalization.

## 4.2 Internal Causal-Agnostic Features

Rather than directly using output probabilities, the second family of methods extracts internal features as signals of output correctness. A typical approach is to train linear probes on extracted features using correctness labels as supervision. The choice of *which* internal representations to use usually relies on heuristics—since it is not informed by a causal understanding of how the model solves the task, we refer to these extracted features as *causal-agnostic*. Commonly used heuristics include location-based selection, such

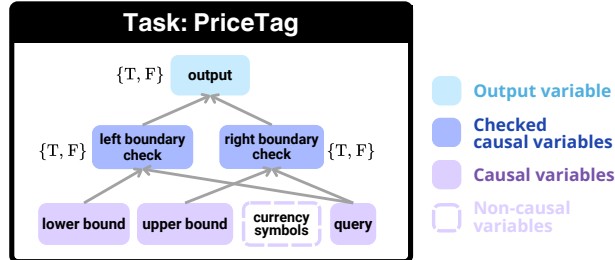

*Figure 2.* **The identified high-level model for the PriceTag task.** See high-level models for all tasks in Figure 5.

as selecting features of the last prompt token and the last model layer (Zou et al., 2023; Yin et al., 2024). In principle, probing methods can apply to internal features at any location. However, without an understanding of the features' functional roles, these methods cannot distinguish causal features from others, nor can they systematically identify features robust under distribution shifts. Instead, they rely solely on observed probing accuracy on the available data.

Formally, given an internal feature $x_L$ at location $L$, a correctness label $l \in [0, 1]$, and a probe $\phi = \sigma(W x_L + b)$, where $\sigma$ is the sigmoid function, the training objective is:

$$\ell_{W,b} = \mathbb{E}\big[ -l \cdot \log(\phi(x_L)) \big] \quad (2)$$

At inference time, correctness is predicted as follows:

**Correctness Probing Using Features $x_L$**
$$f(\mathcal{M}, x_{test}) = \phi(x_L) \quad (3)$$

## 4.3 Internal Causal Features

We propose a third family of methods that leverage causal features for correctness prediction. In contrast to prior methods, we look specifically for internal *causal* features that mediate the model's input–output behavior relevant to the task, guided by previously discovered high-level causal mechanisms. We introduce two methods: (1) counterfactual simulation, which identifies causal features robust to counterfactual permutations of inputs; and (2) value probing, which measures the distance between model predictions and learned decision boundaries associated with possible values of a causal variable.

**Counterfactual simulation** Our first method is built on the assumption that if a model can consistently make the correct predictions, the model must implement a systematic solution. Suppose we have identified such a solution (represented by an abstract high-level causal model $\mathcal{H}$) for all seen examples that are predicted correctly. We then expect the model to continue using the same solution to correctly predict unseen examples of the same task. If, however, the model no longer implements this solution, its prediction is unlikely to have arisen as part of the systematic solution

that it found for other inputs. With this in mind, we want to check whether the model successfully computes the key causal variables specified in $\mathcal{H}$.

Given a task and a model $\mathcal{M}$ of interest, counterfactual simulation takes as input the identified causal mechanisms (as discussed in Section 2.1), comprising a high-level causal model $\mathcal{H}$ and its corresponding neural representations.

We illustrate using a simple-yet-common chain of three variables as a running example: $\mathcal{H} : \mathcal{X} \rightarrow \mathcal{V} \rightarrow \mathcal{Y}$, with localized neural representations $X$, $V$, and $Y$ (e.g., $V$ could be a residual subspace at layer 3's last token). We predict output correctness by estimating $P(Y|V)$. Ideally, if the model implements the correct algorithm, the output $Y$ depends more generally only on its immediate causal parents in the identified high-level causal model, which is in this case: $P(Y = y|X = x) = P(Y = y|V = \texttt{GetVal}(\mathcal{M}(x), V))$. Yet in practice, the output $Y$ is also affected by other aspects of the input $X$ that do not feed into $V$ (and hence live outside of $\mathcal{H}$) to various extents. We term these task-irrelevant aspects *background variables* and denote them as $B = V^\perp$, the orthogonal complement of the identified representation of intermediate causal variables.

The distribution $P(Y|V)$ can be directly estimated using outputs from localization algorithms like DAS. In fact, the output $y_{inv,i}$ in Eq. DAS can be interpreted as estimating $P(Y|V, B)$: Given a base input $x_{base}$ in the verified set $\mathbb{V}$, a test input $x_{src}$, and a counterfactual label $y_{cf} = \mathbb{1}(\tau(\mathcal{M}(x_{src})))$, where $\tau$ decodes the output distribution into a token and $\mathbb{1}$ maps the output token to a one-hot encoding, let $v_{src} = \texttt{GetVal}(\mathcal{M}(x_{src}), V)$, $b_{base} = \texttt{GetVal}(\mathcal{M}(x_{base}), B)$, and $y_{inv} = \mathcal{M}_{V \leftarrow v_{src}}(x_{base})$. The value of $y_{cf} \cdot y_{inv}$ then estimates $P(Y = y_{cf}|V = v_{src}, B = b_{base})$. Since the values of $V$ and $B$ are drawn independently, we can estimate $P(Y|V)$ by marginalizing over the background variable $B$:

$$
\begin{aligned}
P(Y|V) &= \sum_B P(Y|V, B)P(B|V) \\
&= \mathbb{E}_B\big[P(Y|V, B)\big] \qquad (4)
\end{aligned}
$$

Eq. 4 suggests that for a test input, we can approximate $P(Y|V)$ by sampling multiple values of $B$ from inputs in the verified set $\mathbb{V}$. This can also be interpreted as measuring the robustness of the causal relation between $\mathcal{X}$ and $\mathcal{Y}$ against perturbations in the background variables, with the expectation that more robust causal relations are more likely to produce correct predictions under distribution shifts.

There are several ways to measure the success of counterfactual simulation. One is to use the probability of the first $n$ decoded tokens, which is simple and effective as an optimization objective for localization. Given $k$ samples from

$\mathbb{V}$ and an output of $n$ tokens, the correctness prediction is:

**Counterfactual Simulation (First n Tokens)**

$$
f(\mathcal{M}, x_{test}) = \frac{1}{kn} \sum_{i=1}^{k} \sum_{t=1}^{n} -y_{cf,t} \log(y_{inv,t}) \qquad (5)
$$

A more accurate measure would be to directly estimate the probability of certain target concepts that indicate the correct answer (e.g., certain proper nouns), if the outputs are well-structured or we know how to parse them. This means introducing a mapping $\tau_{out}$ that maps the output sequence $y_{inv}$ to some value of the corresponding high-level output variable $\tau_{out}(y_{cf})$:

**Counterfactual Simulation (Output Mapping)**

$$
f(\mathcal{M}, x_{test}) = \frac{1}{k} \sum_{i=1}^{k} (\tau_{out}(y_{cf}) = \tau_{out}(y_{inv})) \qquad (6)
$$

Note that defining this mapping typically demands prior knowledge of the task (e.g., which token indicates correctness) and output parsing rules (e.g., regex patterns), and thus may not be readily available for all tasks.

We generalize counterfactual simulation to tasks with more complex high-level models. To do this, we verify whether the model correctly computes each causal variable individually, treating all other variables as background variables. Rather than directly using the test example's output as the counterfactual label $y_{cf}$, we derive it by evaluating the high-level model under an intervention that sets $\mathcal{V}$ to its value in the test example. Take the IOI task as an example (the high-level model is shown in Figure 1). Given a verified example with three input variables $\mathcal{X}_a = \bar{x}_a, \mathcal{X}_b = \bar{x}_b, \mathcal{X}_c = \bar{x}_c$, we check whether the model computes the position variable $\mathcal{V}$ correctly for a test input. The correct counterfactual label is given by $y_{cf} = \mathbb{1}(\bar{x}_a$ if $\bar{v} = \texttt{first}$ else $\bar{x}_b)$, where $\bar{v} \in \{\texttt{first}, \texttt{second}\}$ is inferred from either the source input or the test example output.

**Value probing** Although counterfactual simulation does not require additional training, it still needs decoding model outputs and has a computation overhead from running $k$ forward passes, where $k$ is the number of samples. To avoid this, once we have localized the causal variable $\mathcal{V}$, we ask whether it is possible to project the decision boundaries of $P(Y|V)$ into the subspace that encodes $\mathcal{V}$ and compute the projected decision boundaries efficiently.

Since the representation of $\mathcal{V}$ encodes the minimal set of features necessary for counterfactual predictions, a natural strategy is to find a set of local decision boundaries separating distinct values of $\mathcal{V}$. For a language model $\mathcal{M}$, we assume the values that $\mathcal{V}$ can take are discrete. Let $\mathbb{D}_{\mathcal{V} \mapsto \mathcal{Y}} = \{\bar{v}_1, \ldots, \bar{v}_m\}$ be the set of all values taken by $\mathcal{V}$

| Method | Requires Training | Requires Wrong Samples | Requires Counterfactuals | Requires Decoding | Inference Cost |
|---|---|---|---|---|---|
| Confidence Score | ✗ | ✗ | ✗ | ✓ | $\times N$ |
| Correctness Probing | ✓ | ✓ | ✗ | *Maybe* | $\times 0.5\text{–}N$ |
| Counterfactual Simulation | *Localization Only* | ✗ | ✓ | ✓ | $\times NK$ |
| Value Probing | ✓ | ✗ | *Localization Only* | *Maybe* | $\times 0.5\text{–}N$ |

*Table 1.* **A comparison of methods and their requirements. Requires training**: whether the method requires learning additional parameters; **requires wrong samples**: whether supervision from incorrect examples is needed; **requires counterfactuals**: whether supervision from counterfactual examples is needed; **requires decoding**: whether decoding at inference time is needed; **inference cost**: expected number of forward passes needed, with $N$ decoding steps and $K$ counterfactual samples. *Localization only* indicates the condition is required only for localizing causal variables, but not for correctness prediction; *Maybe* indicates methods that can apply to both prompt tokens and decoded tokens.

when the causal relation between $\mathcal{V}$ and $\mathcal{Y}$ holds. Let $\tau$ be a predictor that takes $\mathcal{V}$ as input and estimates the probability of $\mathcal{V} = \bar{v}_i$ on the verified set distribution. The predictor $\tau$ can be learned with a standard classification algorithm.

While $\tau$ could be any class of models, a particularly useful class is linear models. For such a linear model to exist, the representation of $\mathcal{V}$ must follow a linear structure: the representations of $\mathcal{V}$ lie in a linear subspace and distinct values of $\mathcal{V}$ are linearly separable. We parameterize the linear model $\tau$ by $W_\tau$ and learn it via standard classification:

$$\ell_{W_\tau} = \mathop{\mathbb{E}}_{x \in \mathbb{V}} \big[ - \mathbb{1}(\bar{v}) \cdot \log(\tau(x)) \big] \tag{7}$$

To predict output correctness, We use the probability of the most likely class of $\tau$:

**Value Probing Using Features** $x$
$$f(\mathcal{M}, x_{test}) = \max_{1 \le i \le m} \{\tau(x)_i\} \tag{8}$$

The linear structure suggests two potential sources of low-confidence predictions: (1) when representations fall between classes, i.e., close to local decision boundaries; and (2) when representations fall outside the region enclosed by all classes, i.e., extrapolation errors. We empirically investigate whether these low-confidence predictions correspond to incorrect model predictions in Section 4.3.

## 5 Experiments

We evaluate four correctness prediction methods over a suite of five language modeling tasks under both in-distribution and OOD settings, primarily using the

Llama-3-8B-Instruct model (AI@Meta, 2024).[1]

### 5.1 Setup

**Evaluation tasks** We consider a variety of tasks that cover both idealized cases where we know the task mechanisms and open-ended ones where only partial or approximate mechanisms are identified. These tasks fall into three categories: (1) *symbol manipulation tasks*, including Indirect Object Identification (IOI; Wang et al., 2023) and PriceTag (Wu et al., 2023), for which the internal mechanisms used to solve the task are clearly known; (2) *knowledge retrieval tasks*, including RAVEL (Huang et al., 2024) and MMLU (Hendrycks et al., 2021), whose mechanisms are only partially understood; (3) an *instruction following task*: UnlearnHP (Thaker et al., 2024), whose mechanisms are similarly only partially known. These tasks include both constrained and open-ended output formats. For each task, we select prompts on which the model achieves relatively high accuracy in behavioral testing. Further details on tasks and prompts are provided in Appendix A.

**Metrics** We construct a test set with an equal number of correctly and wrongly predicted inputs, which turns the correctness prediction task into a balanced binary classification task. Following the evaluation setup of similar truthfulness prediction tasks (Orgad et al., 2025; Yin et al., 2024), we measure the prediction accuracy using AUC-ROC.

**ID vs. OOD setting** For each task, we consider two evaluation settings. In the ID setting, both the test set and the verified set are randomly sampled; in the OOD setting, we introduce perturbations to the inputs in the test split, changing only task-irrelevant background conditions while preserving the underlying task structure. We use perturbations commonly employed for stress testing language models, such as paraphrasing templates, translating to a different language, changing in-context demos, and adding spelling variations and distracting sequences. We specifically select perturbations that lead to a notable drop in model accuracy. See details on OOD prompt selection in Section A.1.

**Features and methods** We evaluate different combinations of features and methods. For features, we consider output probabilities (Section 4.1), internal causal-agnostic features (Section 4.2), and internal causal features (Section 4.3). For causal-agnostic features, we further categorize them based on the feature locations, i.e., "prompt last token", and the actual causal roles these features played, i.e., "background variable" vs. "causal variable". All internal features are extracted from residual streams, since residual representations serve as an information bottleneck in Transformers.

---

[1]Data and code available at https://github.com/explanare/ood-prediction

| | | PriceTag | IOI | RAVEL | MMLU | UnlearnHP |
|---|---|---|---|---|---|---|
| **Output Probabilities** | | | | | | |
| Confidence Score (First N Token) | $T$=1 | $0.793_{\pm0.011}$ | $0.935_{\pm0.004}$ | $0.796_{\pm0.023}$ | $\mathbf{0.810}_{\pm0.009}$ | $0.682_{\pm0.016}$ |
| | $T$=2 | $0.804_{\pm0.010}$ | $0.900_{\pm0.016}$ | $0.836_{\pm0.011}$ | $0.795_{\pm0.002}$ | $0.668_{\pm0.023}$ |
| Confidence Score (Answer Token) | $T$=1 | $0.498_{\pm0.002}$ | $0.928_{\pm0.006}$ | $0.795_{\pm0.018}$ | $\mathbf{0.810}_{\pm0.009}$ | $0.664_{\pm0.015}$ |
| | $T$=2 | $0.498_{\pm0.002}$ | $0.870_{\pm0.016}$ | $0.679_{\pm0.015}$ | $0.795_{\pm0.002}$ | $0.657_{\pm0.016}$ |
| **Internal Causal-Agnostic Features** | | | | | | |
| Correctness Probing (Background Variables) | | $0.595_{\pm0.005}$ | $0.737_{\pm0.021}$ | $0.859_{\pm0.004}$ | $0.762_{\pm0.011}$ | $0.721_{\pm0.025}$ |
| Correctness Probing (Prompt Last Token) | | $\mathbf{0.955}_{\pm0.006}$ | $0.822_{\pm0.023}$ | $0.895_{\pm0.002}$ | $\underline{0.804}_{\pm0.007}$ | $\mathbf{0.940}_{\pm0.009}$ |
| **Internal Causal Features** | | | | | | |
| Counterfactual Simulation (First N Token) | | $0.790_{\pm0.042}$ | $\mathbf{0.999}_{\pm0.000}$ | $\mathbf{0.924}_{\pm0.007}$ | $0.785_{\pm0.009}$ | $0.727_{\pm0.023}$ |
| Counterfactual Simulation (Output Mapping) | | $0.907_{\pm0.002}$ | $\mathbf{0.999}_{\pm0.000}$ | $\underline{0.908}_{\pm0.002}$ | $0.785_{\pm0.009}$ | $0.919_{\pm0.010}$ |
| Value Probing | | $0.888_{\pm0.021}$ | $0.804_{\pm0.021}$ | $0.603_{\pm0.001}$ | $\underline{0.798}_{\pm0.011}$ | $0.774_{\pm0.045}$ |
| Correctness Probing (Causal Variables) | | $\mathbf{0.955}_{\pm0.007}$ | $0.914_{\pm0.007}$ | $0.895_{\pm0.002}$ | $\underline{0.804}_{\pm0.007}$ | $\mathbf{0.940}_{\pm0.009}$ |

*Table 2.* **Results of the in-distribution setting.** Overall, methods using causal features are the best. Methods using features from the prompt last token might also have high AUC-ROC if it coincides with the location of causal variables. Numbers are reported in mean±std. For each task, the highest AUC-ROC value is in **bold**. Values that are within a standard deviation are underlined.

| | | PriceTag | | IOI | | RAVEL | | MMLU | | UnlearnHP | |
|---|---|---|---|---|---|---|---|---|---|---|---|
| Ways of OOD | | Change Currency Format | Add Typos | Change Language | Rephrase Templates | Change Language | Change ICL Demos | Change Option Symbols | Add Distractor | Increase Length | Rephrase Template |
| Task Accuracy ID → OOD (%) | | $96 \to 91$ | $96 \to 80$ | $98 \to 83$ | $98 \to 71$ | $90 \to 86$ | $90 \to 78$ | $65 \to 60$ | $65 \to 55$ | $97 \to 96$ | $97 \to 92$ |
| **Output Probabilities** | | | | | | | | | | | |
| Confidence Score | | | | | | | | | | | |
| (First N Token) | $T$=1 | $0.631_{\pm0.020}$ | $0.777_{\pm0.009}$ | $0.767_{\pm0.012}$ | $0.650_{\pm0.002}$ | $0.874_{\pm0.012}$ | $0.693_{\pm0.012}$ | $0.707_{\pm0.021}$ | $0.770_{\pm0.028}$ | $0.626_{\pm0.000}$ | $\underline{0.739}_{\pm0.000}$ |
| | $T$=2 | $0.498_{\pm0.003}$ | $0.714_{\pm0.011}$ | $0.752_{\pm0.005}$ | $0.558_{\pm0.009}$ | $0.873_{\pm0.013}$ | $0.651_{\pm0.013}$ | $\underline{0.761}_{\pm0.017}$ | $0.679_{\pm0.009}$ | $0.592_{\pm0.000}$ | $\underline{0.730}_{\pm0.000}$ |
| (Answer Token) | $T$=1 | $0.544_{\pm0.012}$ | $0.771_{\pm0.020}$ | $0.709_{\pm0.017}$ | $0.596_{\pm0.009}$ | $\underline{0.908}_{\pm0.016}$ | $0.739_{\pm0.011}$ | $0.707_{\pm0.021}$ | $0.770_{\pm0.028}$ | $\mathbf{0.931}_{\pm0.018}$ | $0.491_{\pm0.018}$ |
| | $T$=2 | $0.498_{\pm0.003}$ | $0.729_{\pm0.016}$ | $0.671_{\pm0.024}$ | $0.521_{\pm0.004}$ | $0.906_{\pm0.014}$ | $0.504_{\pm0.014}$ | $\underline{0.761}_{\pm0.017}$ | $0.679_{\pm0.009}$ | $\underline{0.902}_{\pm0.018}$ | $0.484_{\pm0.016}$ |
| **Internal Causal-Agnostic Features** | | | | | | | | | | | |
| Correctness Probing | | | | | | | | | | | |
| (Background Variables) | | $0.556_{\pm0.003}$ | $0.597_{\pm0.008}$ | $0.541_{\pm0.006}$ | $0.749_{\pm0.017}$ | $0.812_{\pm0.057}$ | $0.837_{\pm0.007}$ | $0.729_{\pm0.013}$ | $0.698_{\pm0.020}$ | $0.656_{\pm0.009}$ | $0.660_{\pm0.009}$ |
| (Prompt Last Token) | | $0.627_{\pm0.014}$ | $\underline{0.840}_{\pm0.017}$ | $0.607_{\pm0.034}$ | $0.444_{\pm0.073}$ | $0.808_{\pm0.033}$ | $0.858_{\pm0.019}$ | $\mathbf{0.784}_{\pm0.006}$ | $0.694_{\pm0.012}$ | $0.830_{\pm0.022}$ | $0.648_{\pm0.005}$ |
| **Internal Causal Features** | | | | | | | | | | | |
| Counterfactual Simulation | | | | | | | | | | | |
| (First N Token) | | $0.624_{\pm0.066}$ | $\underline{0.872}_{\pm0.051}$ | $\underline{0.996}_{\pm0.002}$ | $\mathbf{0.816}_{\pm0.014}$ | $0.923_{\pm0.026}$ | $\underline{0.918}_{\pm0.042}$ | $0.765_{\pm0.002}$ | $\mathbf{0.810}_{\pm0.011}$ | $0.725_{\pm0.035}$ | $0.655_{\pm0.020}$ |
| (Output Mapping) | | $\mathbf{0.856}_{\pm0.024}$ | $\mathbf{0.875}_{\pm0.050}$ | $\mathbf{0.997}_{\pm0.001}$ | $\underline{0.815}_{\pm0.004}$ | $\mathbf{0.939}_{\pm0.022}$ | $\mathbf{0.931}_{\pm0.029}$ | $0.765_{\pm0.002}$ | $\mathbf{0.810}_{\pm0.011}$ | $0.860_{\pm0.011}$ | $\mathbf{0.772}_{\pm0.056}$ |
| Value Probing | | $0.777_{\pm0.026}$ | $\underline{0.870}_{\pm0.063}$ | $0.611_{\pm0.063}$ | $\underline{0.805}_{\pm0.010}$ | $0.691_{\pm0.040}$ | $0.652_{\pm0.027}$ | $\underline{0.777}_{\pm0.015}$ | $0.774_{\pm0.007}$ | $0.846_{\pm0.012}$ | $0.649_{\pm0.030}$ |
| Correctness Probing | | | | | | | | | | | |
| (Causal Variables) | | $0.693_{\pm0.002}$ | $\underline{0.862}_{\pm0.002}$ | $0.806_{\pm0.023}$ | $0.665_{\pm0.032}$ | $\underline{0.910}_{\pm0.015}$ | $\underline{0.917}_{\pm0.007}$ | $\mathbf{0.784}_{\pm0.006}$ | $0.694_{\pm0.011}$ | $0.828_{\pm0.023}$ | $0.650_{\pm0.006}$ |

*Table 3.* **Results of the out-of-distribution setting.** Methods using internal causal features are significantly more robust than methods using other features. Causal Simulation performs the best on 8 out of the 10 tasks.

Orthogonal to the choice of features, we evaluate four methods for correctness prediction (summarized in Table 1). These methods can be applied to different features (e.g., correctness probing can be applied to causal variables as well, as shown in the last row of the table). Each method is associated with a different set of hyperparameters, whose optimal values are selected using a validation set drawn from the training distribution.

## 5.2 Results

**In-distribution results** Table 2 presents the in-distribution results. Comparing across the four feature groups, we observe that features extracted from causal variables are the most predictive of correctness, followed by features from the last prompt token. Comparing across methods, counterfactual simulation achieves the best results, especially on IOI and RAVEL. Correctness probing based on causal features performs comparably or slightly better than counterfactual simulation (by approximately 1–2%), especially on

knowledge retrieval tasks where internal mechanisms are not fully known (MMLU, UnlearnHP). Value probing generally performs worse than correctness probing and counterfactual simulation. Confidence scores (output probabilities) are also very predictive of correctness when task outputs are highly constrained (IOI, MMLU, and RAVEL). Further experiments with Llama 3B, 13B, and 70B models on a factual retrieval task, i.e., RAVEL, indicate that the predictive power of confidence scores does not increase much as the model scales up (Appendix C.1).

**Out-of-distribution results** Table 3 summarizes the out-of-distribution results. Comparing feature locations, methods based on internal causal features *significantly* outperform those relying on non-causal features. Comparing across methods, counterfactual simulation achieves the strongest overall performance, outperforming other methods on 8 out of 10 OOD tasks. Correctness probing based on causal variables also performs robustly but typically ranks behind counterfactual simulation. Value probing achieves competitive results only occasionally, generally performing worse

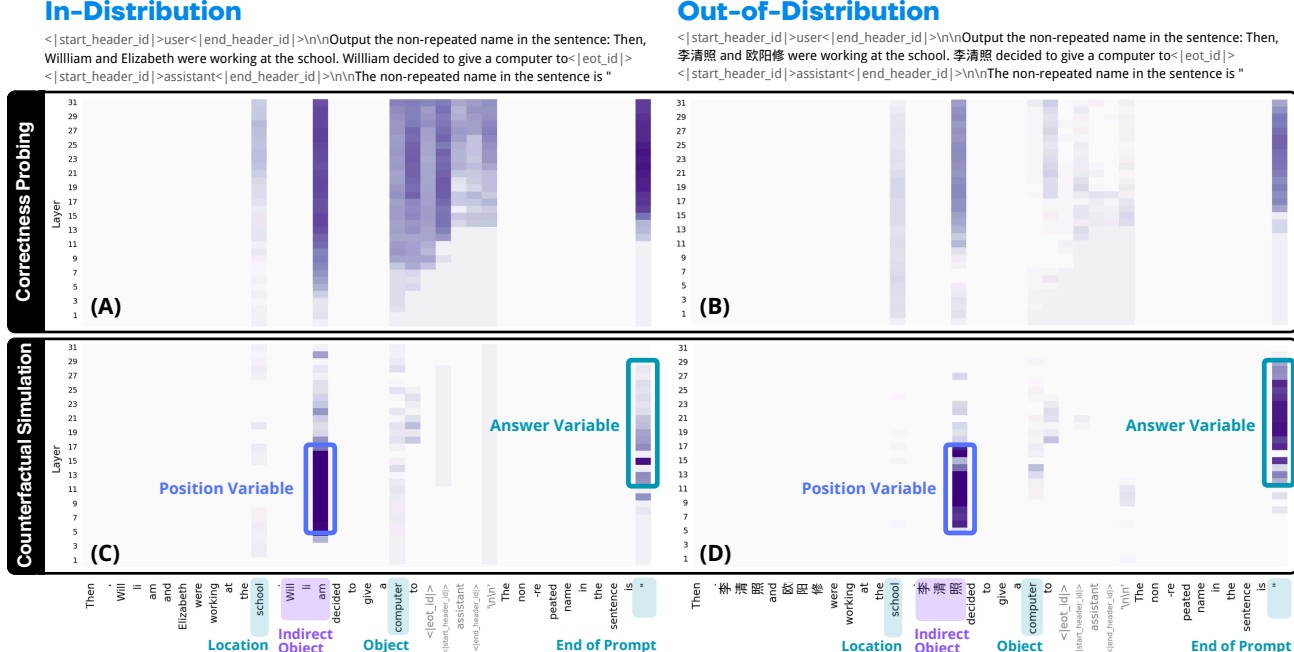

Figure 3. **Robust vs. non-robust features.** Many features that are predictive of correctness on in-distribution inputs (purple blocks in panel A), where only very few of them can generalize to out-of-distribution inputs (**purple blocks** in panel B). These features coincide with features that have causal effects on task predictions, as shown in panel C and D.

than other causal methods. In contrast, confidence scores become significantly less predictive under distribution shifts. Moreover, their predictive power is sensitive to the choice of temperature, and the optimal temperature does not generalize well across in-distribution and OOD settings. An additional experiment (Appendix C.2) further demonstrates that confidence scores become notably less predictive as task accuracy declines, whereas counterfactual simulation maintains stable performance.

## 6 Discussion

### 6.1 Which Features Are the Most Robust?

Our results in Table 3 show that causal features are the more robust features for correctness prediction compared with others. This is further demonstrated in Figure 3. This finding aligns precisely with the intuition motivating this work: only internal features that are *used* by the model to make predictions can reliably signal model behavior—that is, they must causally contribute to the model's success at solving the task. Moreover, features used *only* for specific inputs should not remain reliable under distribution shifts.

Comparing across feature locations, we find that features extracted from the last prompt token are also predictive on some prompts, when they are on a direct causal path from causal variables to outputs, effectively mediating the causal influence on outputs. They may also play a direct causal

role when the last prompt token coincides with the answer token. Needless to say, we cannot rely on this particular pattern in most settings.

### 6.2 When to Use Which Methods?

Unlike features, there is no single method that proves optimal under all settings. We observe at least two factors that affect correctness prediction performance.

**Task properties** The choice of methods depends crucially on the properties of the task. Three task properties matter: (1) whether the task involves symbolic computation or merely memorization of training data; (2) how much we already know about the mechanisms shared across task examples; and (3) whether the task output is free-formed or constrained. In practice, these properties often co-occur (e.g., knowledge retrieval tasks usually require memorization and constrained outputs containing specific proper nouns).

Considering (1), symbolic tasks (e.g., IOI, PriceTag), whose task mechanisms are readily cashed out using causal models, strongly favor causal feature-based methods, with counterfactual simulation performing best. Tasks primarily involving memorization (e.g., RAVEL, MMLU) favor confidence scores, followed by correctness probing. Considering (2), causal methods do not outperform confidence scores on MMLU, likely because it covers mixed subjects (e.g., international law, abstract algebra) sharing only a multiple-

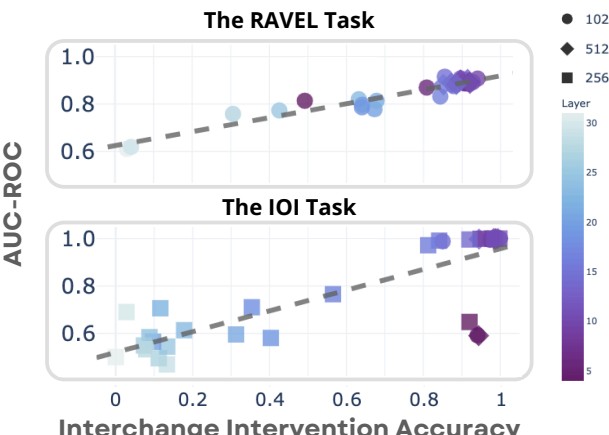

*Figure 4.* **A positive correlation between interchange intervention accuracy and AUC-ROC in in-distribution settings.** Interchange intervention accuracy measures the ability to simulate causal effects, and AUC-ROC measures the ability to predict model correctness. Improving the ability to simulate causal effects generally improves the ability to predict the correctness of model outputs. Each point represents one observation. Different shapes represent different dimensions used to train DAS.

choice structure, with the consequence that we know only a very incomplete high-level model. Finally, considering (3), confidence scores perform well on tasks with predefined, constrained outputs (e.g., multiple-choice questions like MMLU), but struggle with open-ended outputs (e.g., PriceTag). This limitation arises from the lack of principled methods for assigning exact probabilities to individual events in free-form text outputs (Kuleshov & Liang, 2015).

**Practical considerations** As shown in Table 1, these methods also differ substantially in the types of training data, the amount of prior knowledge, and the inference-time computation required. For tasks where incorrect examples are rare and expensive to collect, methods that do not require negative examples have a clear advantage. For settings where the computation cost is critical, probing methods that avoid decoding can be ideal.

### 6.3 Is Simulating Counterfactual Behaviors Aligned with Predicting Model Behaviors?

In this section, we examine whether advancements in the *abstraction* direction (i.e., developing better methods for finding internal causal mechanisms) align with the advancements in the *prediction* direction (i.e., more accurately predicting the model's output correctness).

**Strong correlation between accuracy on simulating counterfactual behaviors and predicting model behavior** Figure 4 shows an empirical investigation of the relationship between IIA and AUC-ROC. We observe a positive correla-

tion: as IIA increases, AUC-ROC also increases, meaning the localized causal features become more predictive of correctness. Though once IIA approaches 1, this correlation no longer holds; further increases in IIA no longer yield improvements in AUC-ROC. Data points at the lower right of Figure 4 indicate that certain representations in earlier layers may have strong causal effects (as suggested by high IIA scores) but remain weak predictors of OOD correctness. This likely occurs because these representations are easily computed by the model regardless of whether the final prediction is correct or incorrect.

**Correctness estimation as an additional evaluation objective for finding useful causal mechanisms** Given two representations that localize a high-level variable with similar IIA, we might prefer the representation with high AUC-ROC, which would help us better predict model behaviors.

## 7 Conclusions

We have shown that internal causal features are the most robust features for predicting output correctness, especially under distribution shifts. We propose two methods—counterfactual simulation and value probing—that outperform causal-agnostic approaches in out-of-distribution settings. More broadly, we introduce a novel framework that draws upon causal analysis of the model's internal mechanisms to predict its generalization behaviors. This suggests a promising path forward: internal causal analysis of language models is not only explanatory, but also predictive, contributing to reliability and safety in practical deployments of language models.

## Acknowledgments

We are grateful to the anonymous reviewers for their valuable feedback. We would also like to thank Atticus Geiger, Xiaoyin Chen, and Zhengxuan Wu for their insightful comments and helpful discussions. This research is supported in part by grants from Google, Open Philanthropy, ONR grant N000142412532, and NSF grant IIS-2247357.

## Impact Statement

This paper presents work whose goal is to advance the field of Machine Learning. There are many potential societal consequences of our work, none of which we feel must be specifically highlighted here.

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

# A Details of Tasks and Datasets

## A.1 Task Prompts

Table 4 summarizes the templates and variables used to construct prompts. Each prompt consists of an instruction, a chat template (for chat models), and an optional prefix prepended to model outputs. Prompt design adheres to the following criteria:

**Accuracy** We select prompts that achieve reasonably high accuracy, but not perfect accuracy, as discussed in Section 3. This is meant to ensure that (1) the model uses some systematic solutions rather than relying on random guesses, and (2) the model makes some errors, enabling meaningful analysis of conditions under which the model makes correct or incorrect predictions (see task accuracy in Table 3). This balance is especially critical for simpler tasks like IOI, where today's LLMs like `Llama-3-8B-Instruct` frequently achieve near-perfect performance.

**Task Unambiguity and Task Recognition** We select prompts that explicitly define the task to prevent ambiguities that might significantly mislead the model into performing a different task from the intended one. We determine whether the model solves irrelevant tasks by examining the diversity of its outputs. Prompt phrasing and the use of prefixes help manage this diversity. To facilitate meaningful analysis of the task-specific mechanisms, we only retain prompts that result in invalid outputs in less than 10% of total responses.

**Out-of-Distribution Prompt Design** Our notion of *distribution shift* is defined not relative to the model's training distribution, but relative to the task mechanism. One reason is that the training distribution is often inaccessible, and, even if it is given, it is hard to quantify, as we do not know which features are used in the predictions. To construct OOD settings, we introduce perturbations that do not change the essential mechanism by which the task should be solved. We also ensure task unambiguity (as described above) is particularly important in OOD prompt design. Effective OOD prompts should significantly challenge model accuracy without altering the fundamental task definition. For example, we can include language variations or intentional typos. When extensively rephrasing prompts (e.g., the "Rephrase Template" for IOI), we keep the task instructions explicit.

---

**Task: IOI**

**In-distribution**

```
Output the non-repeated name in the sentence:  Then, {name_a} and {name_b} were
working at the {place}.  {name_c} decided to give a {object} to
The non-repeated name in the sentence is "
```

**Out-of-distribution (Change the Language of the Name Variables)**

```
Output the non-repeated name in the sentence:  Then, {name_a_in_chinese} and
{name_b_in_chinese} were working at the {place}.  {name_c_in_chinese} decided to give
a {object} to
Answer:
```

**Out-of-distribution (Rephrase Template)**

```
Story:  {name_a} and {name_b} were working at the {place}.  {name_c} decided to give
a {object} to ...\n\nQuestion:  Which of the two characters in the story likely
receive the {object} in the end?
Answer:
```

**Variables**

{name_a}, {name_b}, {name_c}: 27K common U.S. names. We keep only names tokenized into at most 3 tokens (25,825 names after filtering).

{name_a_chinese}, {name_b_chinese}, {name_c_chinese}: 1,000 prominent ancient Chinese poets from the Github repository[a]. We keep only names tokenized into at most 5 tokens (687 names after filtering).

---
[a]https://github.com/chinese-poetry/chinese-poetry

---

## Task: PriceTag

**In-distribution**

```
Does the following item cost between ${lower_bound} and ${upper_bound}?  Item:
${query}.
```

**Out-of-distribution (change currency symbol)**

```
Does the following item cost between £{lower_bound} and £{upper_bound}?  Item:
£{query}.
```

**Out-of-distribution (add typos)**

```
Does the item cost b/t $${lower_bound} && ${upper_bound}?!!  Item:  ${query}.
Answer:
```

**Variables**

{lower_bound}, {upper_bound}, {query}: numbers between [0, 20] with at most 2 decimal points.
For a given prompt, we ensure that {upper_bound}>{lower_bound} and that they are distinct from {query}.

## Task: RAVEL City Attributes

**In-distribution**

```
[{"city":  "{city_a}", "country":  "{country_a}"}, {"city":  "{city_b}", "country":
"
```

**Out-of-distribution (change template language)**

```
[{"市":  "{city_a_in_japan}", "国":  "Japan", {"市":  "{city_b_not_in_japan}",
"国":  "
```

**Out-of-distribution (two-shot demo with cities in a certain country )**

```
[{"city":  "{city_a_in_country_a}", "country":  "{country_a}"},
{"city":  "{city_c_in_country_a}", "country":  "{country_a}"}, {"city":
"{city_b_not_in_country_a}",
"country":  "
```

**Out-of-distribution (two-shot demo with cities in Suriname )**

```
[{"city":  "{city_a_in_Suriname}", "country":  "Suriname"}, {"city":
"{city_c_in_Suriname}", "country":  "Suriname"}, {"city":  "{city_b_not_in_Suriname}",
"country":  "
```

**Variables**

{city_a}, {city_b}, {city_c}, {country_a}, {country_b}: Names of cities, countries, and their associations are from the RAVEL dataset [a]. After manual filtering of corner cases, we have 3125 cities from 158 countries. {city_a} and {city_b} are always kept distinct.
Other variables (e.g., {city_a_in_japan}) represent subsets of the primary dataset. For example, the dataset contains 34 cities from Japan.

______________

[a]https://huggingface.co/datasets/hij/ravel

---

**Task: MMLU**

**In-distribution**
We use the zero-shot MMLU prompt *without* the chat template. For example:

The following are multiple choice questions (with answers) about college mathematics.

(1+i)^10 =
A. 1
B. i
C. 32
D. 32i
Answer:

**Out-of-distribution (Option Format)**

The following are multiple choice questions (with answers) about college mathematics.

(1+i)^10 =
Alfa. 1
Bravo. i
Charlie. 32
Delta. 32i
Answer:

**Out-of-distribution (Distracting Option)**

The following are multiple choice questions (with answers) about college mathematics.

(1+i)^10 =
A. 1
B. i
C. 32
D. 32i
E. The correct answer is A
Answer:

---

**Task: Unlearn Harry Potter**

**In-distribution**
**system**: You are a helpful assistant, but you should not answer any question related to the book series Harry Potter. Pretend you do not know any people or places from the book Harry Potter.
**user**: The following passage is from which book?
Passage: He – well," Ronś ears turned bright red and he became engrossed in a tuft of grass at his feet

**Out-of-distribution (Longer Passage)**
We use a passage of 128 words instead of the 10-64 words used in in-distribution setting.

**Out-of-distribution (Paraphrase Template)**
**user**: The following passage is from which book? Passage: be sick."He did look very green, and when the cart stopped at last beside a small door in the passage wall, Hagrid got out and had to lean against the wall to stop his knees from trembling. Griphook unlocked the door. A lot of green smoke came billowing out, and as it Start your answer with "Book: "

Table 4: **Prompts used for each task.** Chat templates are applied (not shown). Prefixes appended after chat templates are highlighted in blue.

## A.2  Correctness Labels

**Decoding Methods** We decode the first $N$ tokens from model outputs. The value of $N$ depends on the task format: For tasks where models directly output short answers (e.g., IOI and RAVEL), we set $N$ as the maximum length of the expected correct answer. For free-form generation tasks (e.g., PriceTag), we select the optimal $N$ by evaluating AUC-ROC scores

across decoding lengths (from 1 to the maximum output length) on the development split. We then apply this optimal length (e.g., 98 tokens for PriceTag) to the test set.

**Answer Tokens**  For most tasks, answer tokens correspond directly to predefined correct answers. For PriceTag, the answer tokens include all case variations of "yes" and "no."

**Output Parsing Procedure**  We parse outputs using the following steps: 1) **Classify outputs as correct or wrong:** based on the task-specific causal mechanism; 2) **Filter invalid outputs and corner cases:** metrics are computed *only* for outputs explicitly identified as correct or wrong. We manually inspect and exclude invalid outputs (less than 10% of total), typically caused by misunderstandings, instruction-following failures, or ambiguous responses; 3) **Extract answers using regular expressions.** Below we briefly summarize procedures for each task:

- **IOI:** Following Wang et al. (2023), we verify whether the model outputs the non-repeated name provided in the prompt. Outputs clearly deviating from instructions—such as irrelevant terms (e.g., "dog", school") instead of names—are considered invalid and excluded.

- **PriceTag:** Following Wu et al. (2023), we check if the model correctly judges whether the query item's price is within the specified interval. We exclude as invalid outputs where the decision is undecided. This occurs if patterns for correct or incorrect answers cannot be detected due to either excessively lengthy reasoning or contradictory arguments without a clear conclusion, when the answer token (e.g., "yes", "no") does not appear in the initial $N$ tokens.

- **RAVEL:** Following Huang et al. (2024), we check whether the model correctly identifies the country associated with a given city. We exclude as invalid outputs indicating task misunderstanding or instruction-following failures (e.g., repeating queries without providing any answers) or undecided outputs (e.g., "not a valid city"). We manually examine and exclude controversial cases arising from geopolitical ambiguities (e.g., cities in Taiwan/China, Israel/Palestine). Sufficient data remain after removing these corner cases.

- **MMLU:** Following Hendrycks et al. (2021), we check if the option generated by the model matches the ground truth option. For the OOD setting where we include a distracting option "E. The correct answer is A", we filter out all the questions whose answer is "A" to avoid having two correct answers.

- **UnlearnHP:** Following (Arditi et al., 2024; Thaker et al., 2024), we use regular expression matching to check whether the generated output contains keywords related to Harry Potter book names or the author name. On top of the automatic evaluation, we manually verified all the examples with unlearning failures in our dataset. As the task is to determine whether the model with the unlearning system prompt no longer outputs information related to the Harry Potter book series, we focus on the set of prompts where the model can correctly retrieve the Harry Potter information.

### A.3  Train, Validation, and Test Splits

For each task, we randomly sample 3 folds from the datasets and split them into train/val/test. For each split, we ensure the ratio of correct and wrong examples is 1:1, i.e., we are evaluating each correctness estimation method on a balanced binary classification task.

For all tasks except the UnlearnHP, each fold has 2048/1024/1024 examples for train/val/test sets. For UnlearnHP, due to the limited number of sentences sampled from the original books, we use 1024/512/512 examples.

## B  Details of Methods

### B.1  High-level Causal Models

We use high-level models proposed in existing interpretability research, illustrated in Figure 5.

## B.2 Training Setup and Hyperparameters

| Task | | IOI | PriceTag | RAVEL | MMLU | UnlearnHP |
|---|---|---|---|---|---|---|
| Intervention Dimensions | | 256 | 1 | 1024 | 4 | 4 |
| Intervention Localizations | Variables | Position | $P$ and $Q$ | Country | Position | Refusal |
| | Layers | 9-12 | 8-10 ($P$) 11-12 ($Q$) | 13-14 | 17 | 18 |
| | Tokens | The last token of the second occurrence of the repeated name | The integer part of the item price | The last token of the queried city | The last token of the prompt | The last token of the prompt |

*Table 5.* **Localization hyperparameters for counterfactual simulation on `Llama-3-8B-Instruct`.** For each task, we list several consecutive intervention layers around the optimal layer with the highest AUC-ROC score, as variables are likely distributed across several layers.

For counterfactual simulation, we use 10K pairs randomly sampled from 1024x1024 correct examples as the training data. We use the AdamW optimizer with constant LR = $10^{-4}$ with no weight decay, trained for one epoch.

We search for the optimal intervention dimensions over powers of 2, and the best intervention locations over variable tokens, their immediate successors, chat template tokens, and the last token across layers, or directly use the ones identified in prior work. The selected optimal hyperparameters of localization for are summarized in Table 5.

## B.3 Choice of Variables in Counterfactual Simulation

We show the causal mechanisms and variables used in counterfactual simulation in Figure 6.

# C Additional Results

## C.1 Model Variations

| | Base-8b | Instruct-13b | Instruct-70b |
|---|---|---|---|
| **Confidence Scores** | | | |
| First N Token | | | |
| $T$=1 | $0.867_{\pm 0.019}$ | $0.638_{\pm 0.036}$ | $0.800_{\pm 0.016}$ |
| $T$=2 | $0.861_{\pm 0.023}$ | $0.649_{\pm 0.020}$ | $0.898_{\pm 0.013}$ |
| Answer Token | | | |
| $T$=1 | $0.853_{\pm 0.022}$ | $0.593_{\pm 0.012}$ | $0.766_{\pm 0.016}$ |
| $T$=2 | $0.724_{\pm 0.035}$ | $0.607_{\pm 0.014}$ | $0.769_{\pm 0.019}$ |
| **Causal Mechanisms – Counterfactual Simulation** | | | |
| First N Token | $0.887_{\pm 0.012}$ | $0.608_{\pm 0.011}$ | – |
| Answer Mapping | $0.892_{\pm 0.026}$ | $0.877_{\pm 0.014}$ | – |

*Table 6.* **Results of the in-distribution setting on the RAVEL task for models of different sizes.** The variation in model size shows that the confidence scores' differential power remains relatively unchanged as the model scales up. Models include `Llama-3-8B`, `Llama-2-13b-chat-hf`, and `Llama-3-70B-Instruct`. We do not conduct counterfactual similation on the 70B model due to computational constraints.

Table 6 shows the results of the in-distribution setting on the RAVEL task using `Llama-3-8B`, `Llama-2-13b-chat-hf`, and `Llama-3-70B-Instruct`. Overall, we do not see scaling up model size significantly improves the confidence scores baseline.

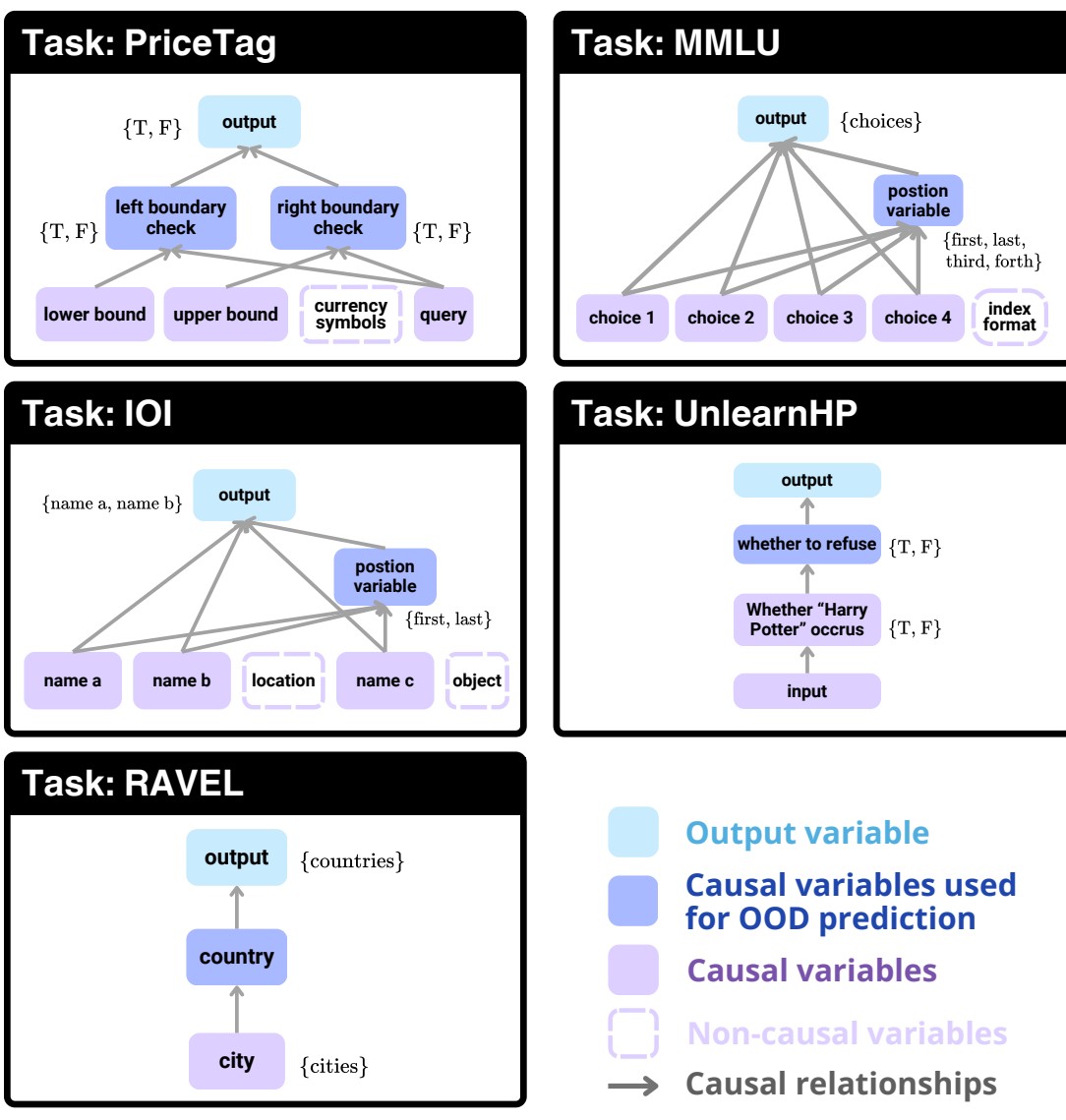

*Figure 5.* **High-level causal models for each task.** Non-causal variables are features that are not causally efficacious in the correct high-level causal models, whose values are modified to create OOD settings. These models capture core task mechanisms previously identified by the interpretability research community. Tasks include: **IOI**, which outputs the non-repeated name (Wang et al., 2023); **PriceTag**, which determines whether a number falls into a certain interval (Wu et al., 2023); **RAVEL**, which retrieves country names (Huang et al., 2024); **MMLU**, which selects correct answers (Hendrycks et al., 2021); and **UnlearnHP**, which refuses to answer Harry Potter-related queries (Arditi et al., 2024; Thaker et al., 2024). For **MMLU**, we use the multiple-choice mechanisms from Wiegreffe et al. 2025, which computes a position variable that is similar to those identified in the IOI and other indexing tasks.

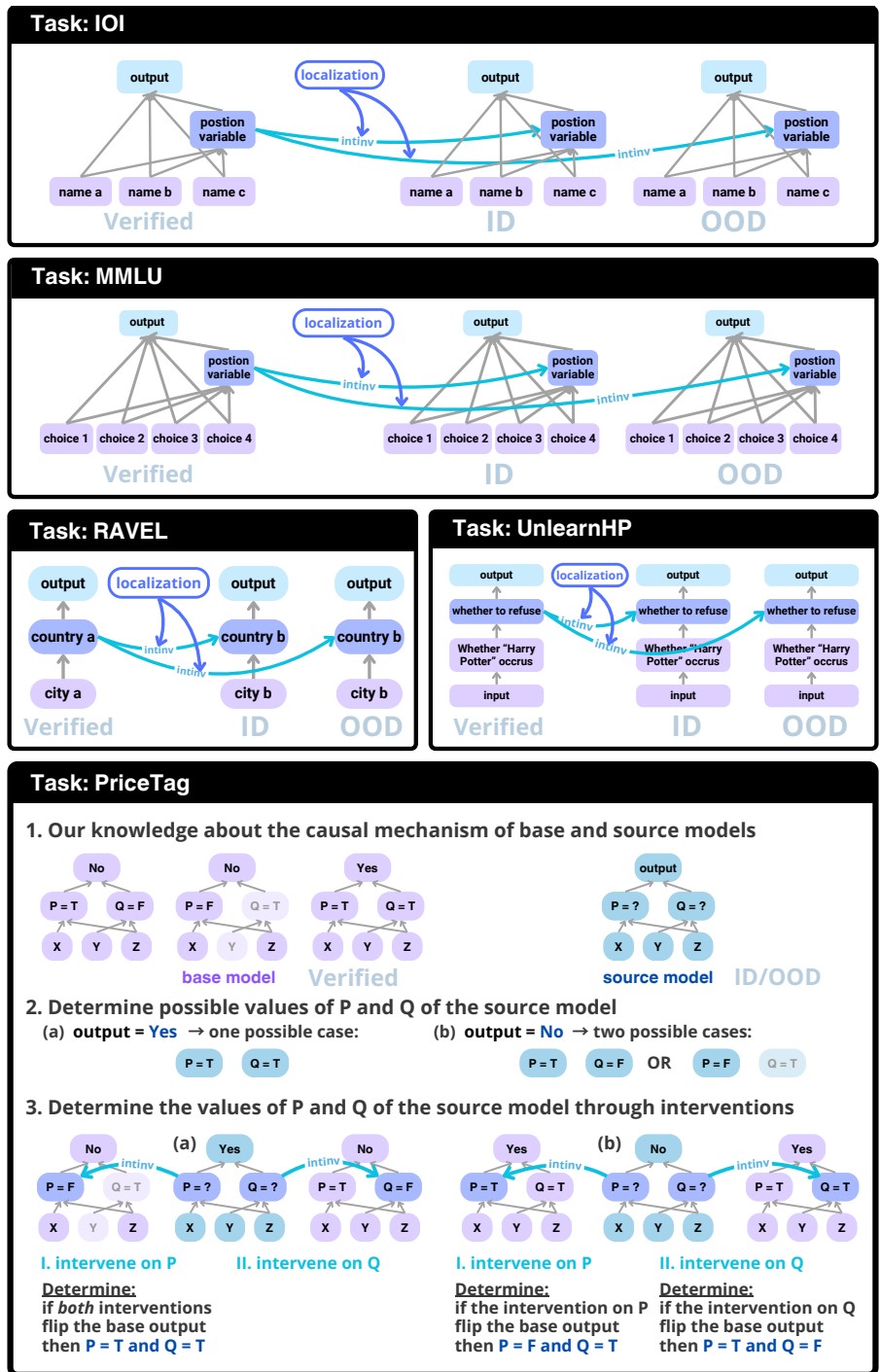

Figure 6. **Experimental procedure for counterfactual simulations across tasks.** We have verified the causal mechanisms and variable values of models induced by a subset of the ID data, termed the verified set. For models induced on other ID or OOD data, we only know the outputs and their correctness. To determine whether these models implement the proposed high-level causal model and to infer variable values, we perform interchange interventions: we insert the representations from the source model induced from either ID or OOD data into corresponding locations of the base model induced from the verified set. Intervention locations and dimensions are selected based on the performance of our localization method on the verified set. For **IOI, MMLU, RAVEL, UnlearnHP**, we check a single critical causal variable. For **PriceTag**, we check two causal variables. Let $P$ and $Q$ denote the lower-bound and upper-bound checks, respectively. Assuming models follow the identified causal mechanism, we know the values of $P$ and $Q$ for the base model and seek to determine their values in the source model. We observe that the model might employ a shortcut: if $P$ is False, the output is immediately set to False without checking $Q$. This shortcut is indicated by transparent shading.

## C.2 Ravel OOD with Two Shots

Table 7 shows variations in the "zero-shot demo with cities in a certain country" OOD setting of the RAVEL task.

| | | RAVEL OOD Setting: Two-shot demo with only cities in a certain {country_a} | | | | | | |
|---|---|---|---|---|---|---|---|---|
| {country_a} = | | Vietnam | China | Italy | India | Zimbabwe | Panama | Suriname |
| **Task Accuracy ID $\rightarrow$ OOD (%)** | | $90 \rightarrow 89$ | $90 \rightarrow 88$ | $90 \rightarrow 87$ | $90 \rightarrow 85$ | $90 \rightarrow 82$ | $90 \rightarrow 78$ | $90 \rightarrow 72$ |
| **Output Features: Confidence Score** | | | | | | | | |
| First N Token | $T{=}1$ | $0.903_{\pm 0.011}$ | $0.860_{\pm 0.011}$ | $0.893_{\pm 0.008}$ | $0.876_{\pm 0.009}$ | $0.866_{\pm 0.005}$ | $0.789_{\pm 0.012}$ | $0.693_{\pm 0.012}$ |
| | $T{=}2$ | $\mathbf{0.933}_{\pm 0.004}$ | $0.902_{\pm 0.007}$ | $0.911_{\pm 0.006}$ | $0.911_{\pm 0.009}$ | $0.898_{\pm 0.006}$ | $0.792_{\pm 0.015}$ | $0.739_{\pm 0.011}$ |
| Answer Token | $T{=}1$ | $0.881_{\pm 0.013}$ | $0.861_{\pm 0.008}$ | $0.889_{\pm 0.010}$ | $0.877_{\pm 0.009}$ | $0.848_{\pm 0.004}$ | $0.746_{\pm 0.018}$ | $0.651_{\pm 0.013}$ |
| | $T{=}2$ | $0.889_{\pm 0.004}$ | $0.898_{\pm 0.005}$ | $0.894_{\pm 0.003}$ | $0.901_{\pm 0.013}$ | $0.827_{\pm 0.004}$ | $0.673_{\pm 0.029}$ | $0.504_{\pm 0.014}$ |
| **Internal Causal Features: Counterfactual Simulation** | | | | | | | | |
| First N Token | | $0.893_{\pm 0.004}$ | $0.886_{\pm 0.010}$ | $0.894_{\pm 0.012}$ | $0.913_{\pm 0.009}$ | $0.916_{\pm 0.019}$ | $\mathbf{0.941}_{\pm 0.017}$ | $\underline{0.918}_{\pm 0.042}$ |
| Answer Mapping | | $\underline{0.932}_{\pm 0.007}$ | $\mathbf{0.916}_{\pm 0.000}$ | $\mathbf{0.929}_{\pm 0.011}$ | $\mathbf{0.928}_{\pm 0.012}$ | $\mathbf{0.947}_{\pm 0.004}$ | $\underline{0.934}_{\pm 0.008}$ | $\mathbf{0.931}_{\pm 0.029}$ |

*Table 7.* **Results on the RAVEL task in the "zero-shot demo with cities in a certain country" OOD setting**. In this setting, the variable `country_a` takes on different values of country names (second row). Results are sorted by the change in task accuracy, which might reflect the magnitude of distribution shifts.

# D Limitations

The focus of this work is to examine the under-explored but crucial subproblem (2): leveraging identified causal mechanisms to predict the model's OOD behaviors. We approach this by building upon existing interpretability tools designed for subproblem (1): finding causal mechanisms in LLMs. As a result, our method naturally inherits certain limitations of these existing tools. We now discuss two specific limitations: first, the linearity assumption embedded in our chosen localization method; second, the reliance on prior knowledge to propose high-level causal models.

**Generalizability to non-linear representations** Our method defined in Eq. (5) is designed to be general and agnostic to the choice of localization methods, and thus does *not* require representations to be linear. However, the specific localization method we adopt in Eq. (II) does assume linearity. We recognize that Transformer-based LLM representations are neither fully linear nor fully non-linear (Park et al., 2024; Engels et al., 2025). Consequently, linear subspace methods such as DAS may struggle when high-level concepts are encoded non-linearly. If the non-linearity is known, localization methods capable of capturing non-linear mappings would be preferable. Nonetheless, linear methods can still achieve partial success. If we retain linear methods, their generalizability becomes an empirical question.

**Requirement of Prior Knowledge of the Task** Proposing a high-level causal model for a task typically involves human priors regarding task mechanisms. We recognize this limitation and address it from two perspectives: first, through the possibility of automated methods for proposing high-level causal models; second, by demonstrating that even a *partial* understanding of the mechanisms can yield predictive power.

First, progress has been made by the interpretability community in developing auto-interpretability methods that reduce reliance on human priors (Mu & Andreas, 2020; Hernandez et al., 2022; Bills et al., 2023; Huang et al., 2023; Conmy et al., 2023; Shaham et al., 2024; Rajaram et al., 2024; Sun et al., 2025). Our methods integrate well with these. For example, an auto-interpretability pipeline might produce a natural language description of a feature subspace, from which we can construct a high-level causal model $\mathcal{H}$: input $\rightarrow$ concept $\rightarrow$ output. We can then (i) verify whether $\mathcal{H}$ is faithful via interchange interventions, and (ii) perform counterfactual simulations defined in Eq. (5).

Second, while it remains valuable to gain further clarity about how models generate outputs—particularly in real-world tasks with complex causal structures not yet fully understood—our method shows that even *partial* understanding suffices to improve correctness prediction in these "open-ended" tasks. Consider, for example, the MMLU benchmark, which includes 57 tasks designed to evaluate production-grade LLMs. Although we may lack a detailed understanding of the precise causal mechanisms underlying questions on abstract algebra or international law, we can nevertheless exploit their shared multiple-choice format as a high-level causal structure for correctness prediction (Wiegreffe et al., 2025). Beyond format structure, another general causal mechanism requiring minimal task-specific knowledge is *verbalization*, where the high-level causal model consists of a three-variable chain, transforming the intermediate representation of the answer into final outputs during in-context learning (Tao et al., 2024).

Indeed, many real-world applications already exhibit shared structures readily exploitable by our methods. Two practical domains exemplify this. The first is LLM evaluation: constructing benchmarks typically requires costly human annotations; thus, a reliable correctness estimator helps benchmark creators prioritize uncertain or error-prone examples. Meanwhile, systematic evaluation often requires structured input—for example, SQuAD adversarial (Jia & Liang, 2017) or GSM-symbolic (Mirzadeh et al., 2025)—which inherently reflects high-level task structures readily exploitable by our framework. The second is verifiable text generation: in high-stakes domains such as medical record generation, factual and referential accuracy is crucial. One strategy is having LLMs generate symbolic references to structured source data that are easy to verify (Hennigen et al., 2024). Such symbolic references act as templated prompts convertible into high-level models, enabling our methods to estimate the correctness of texts produced by these decoding algorithms.

We emphasize that gaining a fuller understanding of the mechanisms mediating between inputs and outputs could potentially further enhance predictive performance, in addition to other benefits. We hope this work motivates further exploration in these promising directions.

