# OpenReview forum: "Internal Causal Mechanisms Robustly Predict Language Model Out-of-Distribution Behaviors"
_ICML.cc/2025/Conference — ICML 2025 poster_

### Official Review · Reviewer_oMv7 · 2025-03-14

**Overall Recommendation:** 4

**Summary:**

Understanding the behavior of complex black-box models has always been a challenge since the inception of deep neural networks (DNNs). This problem has worsened after the introduction of large language models (LLMs). In this paper, the authors investigate whether understanding the internal causal mechanisms of LLMs can improve the prediction of output correctness. In particular, they show that the most robust features for correctness prediction are those that play a causal role in the model's behavior using two proposed methods --
Counterfactual Simulation and Value Probing. The methods are evaluated on diverse tasks like symbol manipulation, knowledge retrieval, and instruction following, where causal features consistently perform better, particularly under out-of-distribution settings. In addition, the methods improve model safety and reliability by providing more accurate correctness estimates.

## update after rebuttal

The authors addressed all my concerns and I vote for the acceptance of this paper.

**Claims And Evidence:**

While the paper provides substantial evidence to support its claims, it lacks some key details that raise questions about the effectiveness of the proposed method.

i) While the methods perform well on the selected tasks, the paper does not extensively address how well these findings generalize to other, more complex real-world language model applications.

ii) The success of the methods relies on identifying causal variables, which may require domain-specific knowledge. It would be great if the authors could comment on the applicability of the approach in scenarios where causal structures are not well understood.

**Essential References Not Discussed:**

NA

**Experimental Designs Or Analyses:**

Yes, I read the experimental setup and thoroughly reviewed the analysis.

**Methods And Evaluation Criteria:**

While the proposed method and evaluation criteria make sense within the context of understanding the internal causal mechanisms of LLMs, there are several questions concerning the methodology and evaluation of the proposed methodology.

i) The benchmark tasks are diverse but they are relatively controlled and lack the full complexity of real-world scenarios (e.g., complex multi-hop reasoning or dialogue systems). This raises concerns about generalizability.

ii) The methods rely on identifying causal subspaces, which may require domain expertise or prior knowledge. This dependency limits the approach’s scalability to more opaque or less understood tasks.

iii) It would be great if the authors could comment on the computational bottleneck of counterfactual simulation, which involves multiple forward passes to assess causal stability. Would this prohibit the use of the proposed method for large-scale language models like GPT-4?

iv) While the paper compares causal vs. non-causal features in Sec. 5.2, a more granular ablation study (e.g., the impact of individual layers or attention heads) could reveal which specific components contribute most to correctness prediction.

v) Will the selection of hyperparameters and causal subspaces for a given task inadvertently lead to overfitting on specific benchmarks?

**Other Comments Or Suggestions:**

NA

**Other Strengths And Weaknesses:**

By leveraging internal causal features rather than relying on output probabilities, one key strength of the proposed method is that it can effectively handle distribution shifts and hallucinations.

**Questions For Authors:**

Please refer to the "Claims And Evidence" and "Methods And Evaluation Criteria" for more details.

**Relation To Broader Scientific Literature:**

The paper presents a novel perspective to understand the correctness of language model predictions.

**Theoretical Claims:**

NA

---

> ### Author Rebuttal · Authors · 2025-04-01
>
> We are grateful that the reviewer found our perspective on correctness prediction novel, and we appreciate the opportunity to discuss the practical value of our work!
>
> ---
> ### Q1. Generalizability to complex real-world tasks
>
> We believe it's helpful to take a step back to examine the source of generalizability concerns and its relation to the broader goal of this work. Specifically, our framework comprises two key components: (1) finding causal mechanisms in LLMs, and (2) leveraging these mechanisms to predict model behaviors, particularly in OOD settings.
>
> **Generalizability questions fall in the design space of problem (1)**, e.g., how to identify the high-level causal model without a strong human prior, how to scale to more complex tasks, etc. These are indeed important problems that mechanistic interpretability community has been focused on. However, even if we could *perfectly* identify the mechanisms underlying every GPT-4 prediction, **it remains an open question whether such mechanisms are actually useful for predicting model behaviors** (Mu et al. 2021; Wang et al. 2023; Sharkey et al. 2025).
>
> Therefore, the focus of this work is to examine this under-explored but crucial step (2). As a result, **we choose to build upon existing tools developed by the interpretability community for problem (1)**. This means we naturally inherit the limitations of these tools.
>
> With this context in mind, we would like to further address limitations of these interpretability tools we used!
>
> **1.1 Reliance on human priors to identify causal mechanisms**
>
> This is indeed a recognized limitation of concept localization methods. We discussed solutions in our response to reviewer uRDA Q3.
>
> **1.2. Scale to real-world complex tasks**
>
> We totally agree that there are many real-world tasks whose complex causal structures are not (yet) fully understood. However, **our method remains applicable when the causal structure is *partially* known, and a surprising number of real-world applications fall into this category.**
>
> In fact, our experiments already include such tasks, e.g., MMLU contains 57 tasks to evaluate production-grade LLMs. While we may not know the exact causal mechanisms behind answering abstract algebra or international law questions, we can still leverage the shared multiple-choice format as high-level structure for correctness prediction.
>
> Below, we highlight two additional practical domains where our methods are directly applicable:
> - **LLM evaluation.** Constructing benchmarks typically requires expensive human annotations. A reliable correctness estimator allows benchmark creators to prioritize uncertain or error-prone examples. Moreover, systematic evaluation often requires structured inputs, e.g., SQuAD adversarial (Jia et al., 2017), GSM-symbolic (Mirzadeh et al., 2024), which inherently reflect high-level task structures that our framework can readily exploit.
> - **Verifiable text generation.** In high-stakes domains like medical records generation, ensuring factual and referential accuracy. One strategy is to have LLMs generate symbolic references to structured source data that are easy to verify (Hennigen 2024). These symbolic references act as templated prompts, which can be converted to high-level models. Our methods allow estimating the correctness of texts generated using these decoding algorithms.
> ### Q2. Computation cost of counterfactual simulation
> We have provided the computation cost in Table 1. Counterfactual simulation is K times more expensive than greedy decoding, where K is the number of counterfactual samples. Empirically, a relatively small set of counterfactual samples, e.g., K=16, is sufficient to achieve high AUC-ROC.
>
> The cost can be further reduced by caching representations before the intervention site. For $Q$ queries from the same task, assume the intervention site is at the middle layer, the average cost per query is $(K+Q+QK)/2Q$, i.e., about $K/2$ times more expensive than greedy decoding when $Q$ is large.
> ### Q3. Which components contribute most to correctness prediction
> Thanks for the suggestion! It indeed aligns with our discussion in Section 6.4 on using correctness estimation as an evaluation task for interpretability methods.
>
> We have experimented with attention heads vs residual streams using GPT-2 on IOI task. The original IOI paper localizes the position variable in the outputs of S-Inhibition heads. These sets of heads produce an AUC-ROC around 0.8, while random heads have an AUC-ROC of 0.5. They underperform residual subspaces, likely because the manually identified heads do not fully capture the causal variable.
> ### Q4. Risk of overfitting causal subspaces
> As with any supervised method, causal subspaces or probes may overfit to the training distribution. This is exactly why we see a slight drop from ID (Table 2) to OOD settings (Table 3) across all tasks/methods. However, **methods using causal features are more robust, i.e. consistently show smaller ID–OOD gaps.**

---

### Official Review · Reviewer_55c3 · 2025-03-14

**Overall Recommendation:** 2

**Summary:**

This paper presents a method for estimating whether a language model’s output is correct by examining “causal” internal representations. It studies both symbolic tasks and open-ended tasks, finding that features which directly mediate model behavior are more reliable than simpler methods. The authors claim that causally grounded features remain robust under prompt changes and distribution shifts.

**Claims And Evidence:**

Yes

**Essential References Not Discussed:**

N/A

**Experimental Designs Or Analyses:**

Yes

**Methods And Evaluation Criteria:**

Yes

**Other Comments Or Suggestions:**

N/A

**Other Strengths And Weaknesses:**

Strengths

1. The authors provide a thorough evaluation across multiple datasets.

2. The paper covers a broad range of tasks under both ID and OOD settings.

Weaknesses

1. The rationale for calling the method “causal” is unclear. The paper never formally defines “causality,” despite it being central to the proposed approach.

2. Several core notions—causal features, causal mechanisms, causal interpretability, and causal role—are mentioned but not defined. The term “causal” appears throughout without a clear theoretical foundation.

3. Some terms, such as “causal representation” and “causal structure,” have precise definitions in existing research. However, the paper neither references nor appears to adopt those definitions. Specifically, the conclusion highlights the assessment and use of “causal representation” and “causal structure” as central contributions, yet neither term is introduced in the main text. In the literature of causal representation learning, a “causal representation” refers to latent variables and the structure among them, while in causal discovery, a “causal structure” is the causal graph whose edges denote structural equations between variables.

**Questions For Authors:**

1. What are the precise definitions of these terms related to causality?

**Relation To Broader Scientific Literature:**

It is well-aligned with work on correctness estimation and the causal mechanism in language models.

**Theoretical Claims:**

N/A

---

> ### Author Rebuttal · Authors · 2025-04-01
>
> We very much appreciate the reviewer’s suggestion to clarify what is distinctively *causal* about our methods, and also how what we are doing relates to important neighboring areas of search, like **causal discovery** and **causal representation learning (CRL)**.
>
> We see this as an opportunity to further clarify the nature of our contribution using the additional space we would get in our next version.
>
> ---
> ### Q1. Clarification on the definition of  “causal”
>
> Our paper is grounded in the literature on **mechanistic interpretability**, and more specifically in techniques that perform interventions on networks to understand how they represent data and make predictions. This is in many respects easier than causal discovery or CRL, since the network is a closed deterministic system.
>
> Nonetheless, the field does not yet have clarity on how networks make predictions, and approaches like ours are seeking to address that limitation. (For additional details on causal abstraction, the family of techniques we draw on, please see our response to **reviewer uRDA Q2.4**).
>
> As we agree that using some of these terms which have technical meanings in important literature (especially “causal representation”) may be confusing to many readers, we intend to reword our summaries of the paper’s main contributions. In particular, we will replace the mention of “causal representation” in the text “assessing stability of causal representations” to “assessing the ability to simulate counterfactual behaviors under causal interventions”. This strikes us as a chance to clarify the diverse ways that causal methods are contributing to AI right now.

---

> > ### Comment · Reviewer_55c3 · 2025-04-04
> >
> > Thank you for your response. However, my concerns regarding the definitions of key causality-related terms remain unaddressed. As mentioned in Q2 of my review, could you please explicitly clarify the definitions of the following terms within the manuscript?
> >
> >   - Causal features
> >
> >   - Causal mechanisms
> >
> >   - Causal interpretability
> >
> >   - Causal role
> >
> > I appreciate your emphasis on mechanistic interpretability, but I’d like to note that it is conceptually distinct from causality: the former typically concerns understanding model predictions (as you mentioned), while the latter centers on intervention and counterfactual reasoning. In my view, if the manuscript's focus is on mechanistic interpretability, it would benefit from significant restructuring, as the current framing and terminology are largely couched in causal language (and the term "mechanistic interpretability" does not appear in the paper).
> >
> > If the intended focus is indeed causality, then I believe it's essential to define the relevant terms precisely to avoid conflating correlation and causation at a conceptual level. Clarifying this would significantly strengthen the clarity and impact of the work.

---

> > > ### Author Response · Authors · 2025-04-04
> > >
> > > We are grateful to the reviewer for their continued engagement with our work!
> > >
> > > - **Causal**: Our use of the word “causal” is the one adopted in standard texts on the subject, e.g., Pearl 2009, Peters et al. 2017, and so on.
> > > - **Causal structure**: What might be less familiar is the type of causal structure we are dealing with. Instead of investigating *partially unobserved* causal structures in the world (as is typical in economics, biology, epidemiology, and so on), we are interested in the causal structure of **a (trained) neural network**. There is a sense in which we have the ground truth for this causal structure: we know exactly what **structural model** characterizes this system.
> > >     - Its **variables** are the neurons in the network and the **functional mechanisms** are given by weight matrix multiplications, non-linear transformations, and so on, across the layers that comprise the neural network.
> > >
> > > The work in mechanistic interpretation on which we are building asks the simple question: is there a **more abstract** causal model that adequately captures the structure of the network when it comes to a particular task that the network successfully performs?
> > >
> > > There is a growing literature within the study of causality concerned with this general question of *when one causal model can be said to **abstract** another.* Answering such questions involves precisely the notions the reviewer identified: interventions, counterfactuals, etc., all understood in the standard way in the field (as in the texts mentioned above). A paradigmatic example of such work on **causal abstraction** is, e.g., this paper by Rubinstein, Weichwald, et al.: https://arxiv.org/abs/1707.00819. In our original submission, we referred directly only to the work in mechanistic interpretability that invokes this subarea of causality research.
> > >
> > > But we would gladly include further clarification of how that line of work relates to the broader field of causality, including references to a broader array of literature, if the reviewer feels that would help allay unnecessary confusion.

---

### Official Review · Reviewer_D6Cz · 2025-03-21

**Overall Recommendation:** 4

**Summary:**

This paper focuses on correctness prediction of large language models. It separates internal features into causal features and background features and suggests two approaches for predicting the correctness of model outputs. In one, permutations are used to determine whether predictions are robust against changes in non-causal features. Another one learns a linear model and checks how close predictions are to the decision boundary.
## update after rebuttal
The authors addressed my concerns and clarified some misunderstandings. Therefore, I have raised my score to accept.

**Claims And Evidence:**

The paper claims that causal representations are beneficial for correctness prediction. There is experimental evidence to support this claim.
It is not fully clear how much these features model meaningful causal relations; however, they improve performance in the experimental evaluation. There is no comparison to any other approach for correctness prediction.

**Essential References Not Discussed:**

There are no essential related works missing that I am aware of.

**Experimental Designs Or Analyses:**

The experimental design is sound and valid.

**Methods And Evaluation Criteria:**

The evaluation makes sense. However, additional experiments would be beneficial to further support the claims made in this paper (see "Claims and Evidence"). There is also no comparison to any other methods. Further information on the experimental setup would be helpful, such as the number of samples considered for each experiment.

**Other Comments Or Suggestions:**

- There is no reference to Figure 1 in the text.
- The evaluation (Section 3.2) should be placed after the methodology and at the start of section 5 since they are not essential to understand the methodology but only matter for the experimental evaluation.
- Should $x$ in Equation 3 not be $x_L$?
- "Since the model’s behavior can vary widely as the input distribution shifts, a reliable correctness prediction method should be able to robust under behavior changes." There is a small grammar mistake in here (Section 3.2).

**Other Strengths And Weaknesses:**

**Strengths**

The idea of using causal features for correctness prediction is promising since predictions based on non-causal features should be less reliable. This should, in particular, help when making predictions on out-of-distribution data, when predictions based on non-causal features that were informative before fail to be useful. The mathematical formalization of the problem is good. Considering output features, last prompt token features, and internal background features as alternatives to the causal features in the experimental evaluation helps highlight the benefit of the proposed approach.

**Weaknesses**

For weaknesses regarding the experimental evaluation, see the fields above.

Clarity and Presentation: The paper illustrates the methodology sufficiently well to understand their approach. However, there is also much room for improvement to make understanding easier. For one, the paper would benefit from a figure on the methodology, outlining the approach in a visually understandable manner. Adding intuition on steps in the methodology would also be helpful. In particular, what do the background variables represent? And what is the causal graph that is assumed by the authors? I understand that $X \rightarrow X_\mathcal{T} \rightarrow Y$, but where does $\mathcal{B}$ fit in? To me, the most sense considering the methodology would be that $X \leftarrow \mathcal{B} \rightarrow Y$ (B acting as a kind of confounding feature), but this is not entirely clear from the paper. Concepts such as distributed alignment search and interchange intervention accuracy would also benefit from more details in this paper to reduce the reliance of readers to be familiar with the corresponding papers.

**Questions For Authors:**

1. What is the intuition and what are the causal assumptions regarding the background variables? What are examples of what should be causal and what background?
2. Can the causal and background variables found by the method be analyzed such that they are understandable for humans?
3. How exactly is the balanced dataset constructed?
4. Under "Value probing", the paper states "...when the causal relation between $\mathcal{X}_\mathcal{T}$ and $\mathcal{Y}$ holds." What does this mean, i.e., when does the causal relation hold and when does it not hold?
5. Why does the counterfactual dataset only consist of samples where the model prediction is correct? Is this sufficient to learn to predict correctness? Why?

**Relation To Broader Scientific Literature:**

To the best of my knowledge, this is the first paper that considers correctness prediction in large language models under a causal perspective. Both the idea and implementation make sense, and the results look promising. However, there is no experimental comparison of other approaches for correctness prediction and how this approach would compare to them.

**Theoretical Claims:**

There are no theoretical claims.

---

> ### Author Rebuttal · Authors · 2025-04-01
>
> We thank the reviewer for the thoughtful feedback! We are especially glad that the reviewer recognizes our work as the first to address correctness prediction in LLMs from a causal perspective.
>
> ---
> ---
> ### Q1. Comparison with strong baselines in correctness estimation
> A major concern is that
> > there is no comparison to any other approach for correctness prediction.
>
> **We respectfully disagree with this characterization, as we have compared with two of the strongest baselines in the literature: confidence scores and correctness probing**. We do realize that these methods are not explicitly labeled as “baselines” in our result tables, which may have caused some confusion. We will revise the manuscript to make the baseline labels more prominent.
>
> To recap our baselines, we have reviewed existing methods (L160) in Section 4.1 (**Confidence Scores with Temperature Scaling**) and Section 4.2 (**Correctness Probing**). These baselines were chosen because they are *widely adopted and representative of state-of-the-art techniques* in correctness estimation (see Section 2.1 for the line of work, and also Kadavath et al., 2022; OpenAI 2023; Chen et al 2024; Orgad et al 2025 for the most relevant SOTA). We have reported their performance in Table 2 and Table 3 in Section 5.2.
>
> ---
> ### Q2. Presentation: Clarification on background variables
> **2.1 Where does the background variable fit into the causal graph**
>
> Please see our responses to **reviewer uRDA 2.1 and 2.2**
>
> **2.2 Are causal and background variables understandable for humans**
>
> This is a valuable question from the explainability perspective–namely, how to explain complex models to humans. In our experiment, **all causal variables correspond to human-interpretable concepts**, as these high-level models were manually specified by interpretability researchers. However, we want to emphasize that **our approach to correctness prediction does not require the causal variables to be human-interpretable**. The causal role of a variable in predicting model behavior is sufficient, even if it is not directly intelligible.
>
> ---
> ### Q3. Details on dataset construction
> - For details on **split generation**, please see our response to **reviewer w7xm Q1**.
> - For **label balancing**, we perform stratified sampling: we first partition all prompts into two groups based on the correctness of the target model predictions. We then randomly sample 1024/512/512 examples from each group.
>
> ---
> ### Q4. Clarification: when does the causal relation $X_T → Y$ hold and when does it not hold
> As discussed in our response to **reviewer uRDA 2.1**, the causal relation $X_T → Y$ is an abstraction of low-level neural networks.
>
> **This abstraction is faithful when $X_T$​ mediates the model’s behavior.** However, neural networks often fail to generalize beyond their training distributions, where the model’s actual behaviors might diverge from what the high-level causal model $H$ predicts. **When this occurs, we say the high-level causal relation $X_T → Y$ no longer holds**, i.e., it is no longer a faithful abstraction of the low-level neural network model. These cases correspond precisely to our out-of-distribution settings.
>
> ---
> ### Q5. Clarification: Why does the counterfactual dataset consist only of correctly predicted samples
> This is indeed one of the interesting findings from our work: **it is sufficient to learn a strong correctness predictor using only correctly predicted examples.** This finding might make more sense if we consider confidence score methods, where correctness predictions are made using the probabilities output alone without using any additional samples.
>
> The question of why Counterfactual Simulation works is in fact tied to the core question we are asking in this work: does understanding the internal causal mechanisms allow us to better predict model behaviors, especially under distribution shift?
>
> Intuitively, if we know that (1) for all correctly predicted examples, the model behaves according to a high-level causal model $H$ (i.e. implements a systematic solution), and (2) for an unseen example, the model does not implement the same solution, **then it is very likely that the model is predicting something abnormal and so the prediction is likely wrong**. Mathematically, this intuition is formalized in Eq 7-9, where the counterfactual simulation measures whether the model implements the same solution as we have observed on the correct samples.
>
> ---
> ### Q6. Presentation
> We appreciate the reviewer’s suggestion to clarify the definitions of "Distributed Alignment Search" and "Interchange Intervention Accuracy."
>
> We will include preliminary explanations of these terms in the revised manuscript!

---

### Official Review · Reviewer_uRDA · 2025-03-22

**Overall Recommendation:** 4

**Summary:**

This paper investigates the use of internal causal mechanisms within language models (LMs) to predict the correctness of their outputs. Rather than relying on traditional confidence scores or heuristic probing of internal activations, the authors propose two methods grounded in causal interpretability, Counterfactual Simulation and Value Probing.

These methods are evaluated across a diverse suite of tasks under both in-distribution and OOD settings. The authors demonstrate that causal features yield more robust correctness estimates than non-causal baselines, particularly under distribution shifts.

The work builds upon the causal abstraction framework and introduces a correctness estimation benchmark using known causal variables in some tasks.

**Claims And Evidence:**

The central claim of the paper is that **internal causal mechanisms are more robust predictors of correctness than non-causal heuristics**, particularly under distribution shift. This claim is supported by:
- Comprehensive experiments across five tasks and ten OOD settings (Tables 2 & 3).
- Comparisons between causal and non-causal features (confidence scores, probing) under consistent evaluation metrics (AUC-ROC).
- Correlations between Interchange Intervention Accuracy (IIA) and correctness AUC (Figure 2), suggesting a link between causal alignment and predictive robustness.

However, one problematic claim is the general applicability of the linear decomposition assumption (Eq. 5). The paper acknowledges limitations but does not provide direct evidence or diagnostics when this assumption fails (e.g., in tasks with entangled representations).

**Essential References Not Discussed:**

NA

**Experimental Designs Or Analyses:**

Overall solid, tasks cover different scenarios.
Why value probing not performing well in a lot scenarios?

**Methods And Evaluation Criteria:**

Counterfactual simulation and value probing are appropriate for the problem of correctness estimation. The use of interchange interventions to isolate causal variables is well-motivated by prior interpretability literature (e.g., Geiger et al., 2021, 2024).

Each task is evaluated under both in-distribution and OOD prompts, with OOD shifts chosen to stress test causal robustness.

**Other Comments Or Suggestions:**

Page 5:

> “We empirically study whether these low-confidence predictions correspond to incorrect predictions in Section 4.3.”

“Section 4.3” may be a typo. Based on context, the relevant content likely appears in Section 5 (Experiments). Please confirm and update accordingly.

Page 5:
> “We evaluate four correctness predictions methods over a suit of five language modeling tasks.”

“We evaluate four correctness prediction methods over a **suite** of five language modeling tasks.”

---

Explicitly present the assumed structural causal model (SCM) as a diagram or formal figure.
The implicit model appears to be:
$X \rightarrow X_T \rightarrow Y, X \rightarrow B \rightarrow Y$

**Other Strengths And Weaknesses:**

Strengths
- Strong empirical benchmark with real OOD variation
- Introduces novel causal predictors (simulation + value probing)
- Bridges interpretability and evaluation

Weaknesses
- Causal assumptions (e.g., disentanglement, uniqueness of $X_T$) not deeply validated
- Theoretical work directly based on prior work (DAS)
- Method assumes knowledge of task structure to define $X_T$

**Questions For Authors:**

See comments above

**Relation To Broader Scientific Literature:**

The paper is directly based on the causal abstraction and mechanistic interpretability literature:
- Geiger et al. (2021, 2024a, 2024b) on DAS and interchange interventions
- Wu et al. (2023), Huang et al. (2024), and others on task-specific circuit discovery
- Work on LLM trustworthiness via internal probes (e.g., Azaria & Mitchell, 2023; Ferrando et al., 2024)

Novelties including:
- Reframing causal representations as predictors of correctness, not just explanatory tools
- Demonstrating improved robustness under shift over confidence scores and surface-level probes

**Theoretical Claims:**

The key theoretical claim is that internal model representations can be linearly decomposed into task-relevant (causal) and background (nuisance) components, enabling causal inference via projection.

This is not formally proven in the paper. The decomposition in Eq. (5) is derived from prior work on causal abstraction and DAS, but:
- No formal conditions for identifiability are given.
- No proofs of convergence or uniqueness of the causal basis Q.
- The assumption that task behavior is mediated by a single variable X_T is strong and underspecified.
- No formal causal graph given.

---

> ### Author Rebuttal · Authors · 2025-04-01
>
> We thank the reviewer for their insightful comments! We are encouraged that they recognized the novelty of our methods, the strength of our empirical results, and the significance of bridging the explanatory and predictive aspects of interpretability analysis.
>
> ---
> ---
> ### Q1. Clarification on the linear decomposition assumption
> Linearity is indeed an assumption made in DAS. We have discussed its implications in the presence of non-linearity and potential remedies in our response to **reviewer w7xm 2.4** and offer additional evidence in **Q4** below.
>
> ---
> ### Q2. Clarification on the causal assumptions and causal model
> **2.1 Causal structures in causal abstraction**
>
> We also appreciate the reviewer’s request to clarify the causal aspects of our approach, which we recognize does rely on previous work on causal abstraction. **There are in fact several causal structures at play, and causal abstraction is about the relationship between different causal structures.** Most basically, there is the causal structure of the neural network itself: a sequence of layers of variables with each layer depending functionally on the previous. But there is also a second causal structure given by the “high level” model $H$ that, in our running example, is given by just three variables, $X, X_T$, and $Y$. **The variable B does not occur in $H$. Moreover, $X_T$ is not unique in the sense that we can have different high-level models representing different levels of abstraction.**
>
> DAS then involves searching for $X_T$ somewhere in the network, such that the interchange interventions are successful. When that happens, we say that H is an approximate abstraction of the network, in the sense that $X_T$ mediates the transformation from task $T$ input to output. In the limit, DAS is guaranteed to find this encoding of $V_T$, provided it exists. **But note that basis $Q$ is not guaranteed to be unique.**
>
> **2.2 The background variable $B$**
>
> With this much, we can say exactly what $B$ is. $B$ is just the orthogonal complement of the identified representation of $X_T$ (e.g., somewhere in the residual stream). It lives outside the simple high-level structure $H$ and is determined by the structure of the network, namely how it encodes $X_T$. **We call it a background variable because it encompasses everything about the input that does not feed into $X_T$** (and ultimately determine the output $Y$).
>
> **2.3 Identifiability** Please see our response to **reviewer w7xm 2.3.**
>
> **2.4 Modeling more complex task structures**
>
> We totally agree that this three-variable high-level structure can be overly restrictive. Indeed, **most tasks studied in our paper involve significantly more complex structures.** For instance, in the IOI task, at least 6 causal variables are involved (Figure 2, Wang et al. 2023), including 3 input variables (IO, S1, S2), 2 position variables (outputs of the Induction and the S-Inhibition Heads), and the output. We use the second position variable as $X_T$ to perform counterfactual simulation. We will clarify in the revised manuscript that the three-variable model is for *illustrative purposes*, and our methodology generalize to more complex settings as demonstrated in our experiments.
>
> ---
> ### Q3. Method assumes knowledge of task structure to define $X_T$
> This is a valid concern, and indeed a known limitation. However, the interpretability community has proposed auto-interp methods to reduce the reliance on human priors (Mu et al. 2021; Hernandez et al., 2022; Bills et al., 2023; Conmy et al., 2023; Rajaram et al., 2024).
> Our proposed methods work well with these methods.
>
> For example, an auto-interp pipeline produces a natural language description of a feature subspace, which can be translated into a high-level causal model $H$: input → concept → output. We can then (i) verify whether $H$ is faithful via interchange interventions, and (ii) use $H$ to perform the counterfactual simulation defined in Eq (7).
>
> ---
> ### Q4. Reasons for why value probing underperform in many settings
> We hypothesize two reasons:
> - **Incomplete coverage of the causal pathway.** Unlike Counterfactual Simulation, value probing does not cover the full causal mechanism from X to Y and thus might fail to detect errors in the $X_T→Y$ computation.
> - **Complexity of decision boundaries.** Linear probe requires not only that the variable can be encoded in a linear subspace, but also that the individual values of variables be linearly separable (L222-225)--a stronger assumption than DAS’s (Eq. 5). Non-linearity and high-dimensional variables generally make the geometry of the decision boundaries more complex and harder to learn. A great example is the country variable in RAVEL, which has over 200 unique values. Although the most frequent countries are linearly separable (as visualized via PCA), the long tail distribution makes learning the complete decision boundaries challenging. This likely explains Value Probing’s underperformance on RAVEL.

---

> > ### Comment · Reviewer_uRDA · 2025-04-03
> >
> > I thank the authors for clarifying my concerns. Therefore I raise my score.

---

### Official Review · Reviewer_w7xm · 2025-03-24

**Overall Recommendation:** 4

**Summary:**

For this paper the authors try to identify internal features of LLM that mediate causal effects on the final prediction output. In settings where LLM predictions align with the actual real-world causal process, causal features are assumed to resemble the ground-truth mechanism and, therefore, allow for predictions which are invariant to external disturbances or out-of-distribution queries.

Towards the goal of identifying causal features, the authors present two methods which work on finding features which are invariant under counterfactual inputs ('counterfactual simulation') or learn a decision boundary from the extracted activations ('value probing'). In both cases the authors first localize features that model an intermediate variable by finding a minimal linearized subspace and decompose this into causally relevant and background variables.

Experiments are conducted over several datasets symbolic manipulation, retrieval, instruction following inspecting AUC-ROC for in-distribution tasks and robustness under OOD settings. Through their experiments the authors find that their proposed methods are capable to identify causal features relevant to the given tasks. This results in high AUC-ROC values for in-distribution settings and superior prediction performance in OOD settings, compared to prior baselines.



Please note, this is an emergency review.



**Update after rebuttal.** In their rebuttal the authors were able to further clarify implications and limitations of their made linearity assumptions in their work. While I acknowledge possible concerns regarding the formal handling of causality, causal representations and structure, I find the concepts to be sufficiently handled for this type of work, and the presented results of figs. 1 and 2 convincing and in line with already existing literature (e.g. the particular positions of decision making within LLMs). I therefore remain with my recommendation to accept the paper.

**Claims And Evidence:**

The authors claim that their proposed methods are able to identify internal model features that are causally relevant to predict the final outcome of the model. To this, the authors conjecture that a linear decomposition of layer residuals into subspaces of causally relevant and irrelevant background features yields the desired representations. While linearity assumptions might not hold under all settings, extensive prior work on feature space linearization exists, supporting the presented claims. From the presented evidence in Figures 1, 2 and tables 2 and 3 it can be furthermore concluded that the presented methods indeed identify causally relevant features as OOD performance remains stable and the visualized identified feature activations align well with the predicted outcome.

**Essential References Not Discussed:**

The authors general motivate and embed their presented work well within the existing literature. The presented work builds on a series of prior work on causal feature extraction from LLM by Geiger et al. . The authors expect extensive knowledge on this line of work, which hinders comprehension and self-enclosedness of the work. Key concepts on the localization process of finding causal mechanisms in LLM or extracting feature subspaces, such as "distributed alignment search" or evaluation metrics such as the "interchange intervention accuracy", are only briefly referred to. The presented methodology seems to rely on linearity assumptions of latent representations which, however, are only insufficiently discussed. In this regard, the paper might be improved by more explicitly discussing identifiability of model representations [1,2], possibilities of non-linear mechanisms identification or their non-identifiability [3,4] and general (causal) perspectives on linearity and subspaces in LLM activations [5].



[1] Mikolov, T., Yih, W. T., & Zweig, G. (2013, June). Linguistic regularities in continuous space word representations. In Proceedings of the 2013 conference of the north american chapter of the association for computational linguistics: Human language technologies (pp. 746-751).
[2] Park, K., Choe, Y. J., & Veitch, V. (2023). The linear representation hypothesis and the geometry of large language models. arXiv preprint arXiv:2311.03658.
[3] Leemann, T., Kirchhof, M., Rong, Y., Kasneci, E., & Kasneci, G. (2023, July). When are post-hoc conceptual explanations identifiable?. In Uncertainty in Artificial Intelligence (pp. 1207-1218). PMLR.
[4] Friedman, Dan, et al. "Interpretability Illusions in the Generalization of Simplified Models." *International Conference on Machine Learning*. PMLR, 2024
[5] Rajendran, Goutham, et al. "From Causal to Concept-Based Representation Learning." *Advances in Neural Information Processing Systems* 37 (2024): 101250-101296.

**Experimental Designs Or Analyses:**

The authors describe the overall experimental setup and use of metrics well and seem setup conduct a proper evaluation. The paper, however, severely lacks in terms of experimental details with regard to the training setup of the proposed method and hyperparameter tuning of prior methods. The authors mention (hyper)parameter optimization and training setup, e.g. for the $\tau$-classifier, but do not specify the exact model setup, optimization method, learning rate or number of samples/iterations. While the proposed methods incorporate additional steps which make reasonable effort to enhance results, there is currently no way of assessing whether or not the evaluation has been setup properly and fair from the current state of the paper.

The evaluation over different datasets, along with results in tables 2 and 3 and the figures 1 and 2, seem to indicate a proper working of the methods with alternating, but constantly better, performance than the compared baselines. The presented visualization generally support the claims of the paper and indicate a good identification of causally relevant features from the internal residual activations for the methods.

**Methods And Evaluation Criteria:**

The specific proposed methods of counterfactual simulation and value probiing are well described and formalized in equations 7-11. The general approach appears to be sound and handles correctly handles implementation of causal considerations for identifying features.

The utilized datasets of Indirect Object Identification, PriceTag, RAVEL, MMLU and UnlearnHP are known or seem to be suited to test the claimed effects. The authors test on an in-distribution and an out-of-distribution setting by altering prompts and utilized concepts. Here, the construction of interventions on the datasets and prompts are well described. The presented prompts are reasonably designed and their rephrased version are suited to evoke OOD behavior by altering sentence phrasing or adding distracting artifacts. Performance is measured in terms of AUC-ROC, task accuracy and interchange intervention accuracy. The individual metrics are suited to assess the respective effects.

**Other Comments Or Suggestions:**

* typo l.120 "desire[d] behavior"

**Other Strengths And Weaknesses:**

**Strengths**

The paper is generally well written and motivated. The presented approaches soundly incorporate the notion of causality and build up existing feature extraction methods. The presented theory and working of the methods is well formalized in the equation.

The identified representation hold desirable properties in terms of robustness and OOD behavior. The authors are able present convincing evidence towards the correct identification of such features.

Finally, the experiments seem to be generally well setup and support the claims made. From the visualizations of identified features the authors are able to demonstrate correct identification of causally relevant factors.



**Weaknesses**

The mentioned weaknesses concern the lack of clarity and insufficient discussion on possible assumptions as mentioned before. Specifically:

1) The lack in clarity on the experimental evaluation makes it impossible to judge the correct setup and comparison of the methods. The authors might want to provide the necessary details, as discussed in the section above.
2) By the decomposition of residual activations in Eq. (5) the authors assume a simplified linear representation of model residual activations. Since the authors are furthermore concerned with causal interactions, they only consider the high-level causal chain of input, intermediates and output. Both assumptions are simplifications to the true working of the model and might lead to a simplified regression towards only the direct parents intermediates of Y. While prior works have shown that linear interpretations might exist, the authors might discuss possible implications and limitations of their approach in setting of more complex causal structures and composed or non-linear behaviour.
3) The authors might improve their paper, by briefly describing the key ideas of the localization process and distributed alignment search, in particular with regard to the necessary assumptions for the utilization of the methods and applicability to the presented setting.

**Questions For Authors:**

Questions mainly concern the weaknesses above:

1) Could the authors provide further details on the experimental setup, such as training parameters and hyper parameter search?
2) Could the authors discuss the required assumptions with respect to linearization of the residual feature space?
3) What are the implications for the methods in terms of expressiveness and variable identifiability of intermediate values which are no direct causes of the final model output? Would the presented methods be able to identify such variables?

**Relation To Broader Scientific Literature:**

LLMs have classically been found to generally struggle with direct causal reasoning [1-5]. Nonetheless, as of today, LLM are utilized in an abundant number of tasks. Recent interest on mechanistic interpretability and circuit extraction techniques pose desirable guarantees in terms of robustness and scalability.  The identification of causally relevant features might help establish particularly strong theoretical quarantees and help scaling and generalization to previously unseen OOD queries.



[1] Jin, Zhijing, et al. "Cladder: Assessing causal reasoning in language models." *Thirty-seventh conference on neural information processing systems*. 2023.
[2] Kıcıman, Emre, et al. "Causal reasoning and large language models: Opening a new frontier for causality." *arXiv preprint arXiv:2305.00050* (2023).
[3] Zečević, Matej, et al. "Causal parrots: Large language models may talk causality but are not causal." *arXiv preprint arXiv:2308.13067* (2023).
[4] Gao, Jinglong, et al. "Is chatgpt a good causal reasoner? a comprehensive evaluation." arXiv preprint arXiv:2305.07375 (2023).
[5] Ashwani, Swagata, et al. "Cause and Effect: Can Large Language Models Truly Understand Causality?." arXiv preprint arXiv:2402.18139 (2024).

**Theoretical Claims:**

Apart from assuming linear decomposability in Eq. (5), the authors present no direct theoretical claims or proofs. Causal relations from inputs variables, intermediate representations to the model output are straight forward and correctly formalized.

---

> ### Author Rebuttal · Authors · 2025-04-01
>
> We thank the reviewer for their detailed feedback! We are glad they found our proposed methods “well formalized” and “soundly incorporate the notion of causality” and appreciated our task design, metrics, and supporting results.
>
> ---
> ### Q1. Experimental Details
> We provide detailed experimental setup below to further assure reviewers that our baseline comparisons are fair.
> - **Datasets**
>     - For each task, we randomly sample 3 folds from all prompts (i.e. the Cartesian product between templates and variable values). Each fold has 2048/1024/1024 examples for train/val/test sets (1024/512/512 for UnlearnHP due to limited source texts). Our dataset sizes are comparable to prior work on truthfulness prediction (e.g., Orgad et al 2025, Gottesman et al 2024).
> - **Training setup and hyperparameters** (selected based on val set accuracy)
>     - **Confidence Score**
>         - Temperature scaling: Search over T = 0.5, 1, 2, 3 and report results for the best two (T = 1, 2).
>         - Output token selection: Search over first-N, N ∈ [1, 100] and answer tokens. Answer tokens are identified via regex.
>         - Aggregation: Experiment with mean (Kadavath et al., 2022; Guerreiro et al., 2023) and min (Varshney et al. 2023), reporting mean  as it outperforms min on most tasks.
>     - **Probing**:
>         - Model: Experiment with `sklearn.linear_layer.LogisticRegression` with default settings (follow Orgad et al. 2025) and `sklearn.svm.LinearSVC` (which has lower accuracy)
>         - Feature location: Search across all layers and tokens.
>     - **Counterfactual simulation**:
>         - Training data: 10K pairs randomly sampled from 1024x1024 correct examples
>         - Intervention dimension: Search over powers of 2; Use 1/256/1024/4/4 for the five tasks.
>         - Intervention location: Use locations identified in prior work or search over variable tokens, their immediate successors, chat template tokens, and the last token across layers.
>         - Optimizer: AdamW with constant LR = 1e-4, no weight decay; trained for one epoch.
> ### Q2. Linearity Assumptions on Representations
> **2.1 Presentation of assumptions**
>
> We fully agree and have made the linearity assumption for value probing explicit in L222-228. We also take this opportunity to clarify the assumptions behind counterfactual simulation below.
>
> **2.2 Counterfactual simulation does not require linearity**
>
> We would like to clarify that Eq. (7) does *not* rely on any assumption of linearity in model representations. The linearity assumption appears only in Eq. (5), which defines *one possible* localization method. Crucially, our formulation in Eq. (7) is designed to be general and agnostic to the choice of localization methods. It remains valid even when using localization methods that operate in non-linear spaces.
>
> **2.3 Causal abstraction assumes access to the full causal structure, and is not subject to the identifiability concerns raised in [3–5]**
>
> The identifiability problems in causal settings arise when some fundamental aspects of the causal structure is unknown, as in the important literature references by the reviewer [3, 4, 5]. By contrast, in our setting, the relevant causal structure--the neural network itself--is assumed to be fully accessible to use. The task of DAS, and related techniques for finding causal abstractions, is simply to check whether a "high-level" causal structure is implicitly implemented in the network. This is not an inference or identification problem, but rather a search problem whose goal is to help us understand how the model represents examples and makes predictions. We will clarify this important distinction at the beginning of the paper.
>
> **2.4 Generalizability to non-linear representations: Transformer representations are not fully linear, yet localization methods like DAS can still have partial success**
>
> We thank the reviewer for raising this point. We agree that the linear subspace approach in Eq. (5) has limitations when high-level concepts are non-linearly encoded. In this case, we can switch to a more suitable localization method (e.g., involving non-linear mappings) and our method in Eq. (7) remains compatible.
>
> If we stick with current linear methods like DAS, generalizability becomes an empirical question. Since Transformer-based LLM representations are neither fully linear nor fully non-linear (Park et al. 2024; Smith 2024; Engels et al. 2024), our results already offer empirical insights—we expect a *partial* success rather than a complete failure with non-linear representations.
> ### Q3. Generalizability to more complex causal structures
> **3.1 Expressiveness** Please see our response to **reviewer uRDA Q2.4**.
>
> **3.2 Variable identifiability** Please see our response to **Q2.3**
> ### Q4. Presentation
> We sincerely thank the reviewer for the suggestions on engaging the broader audience who are less familiar with the interpretability literature. We will add a preliminary on localization methods and evaluation metrics.

---

> > ### Comment · Reviewer_w7xm · 2025-04-03
> >
> > I thank the authors for providing details on the experimental evaluations, which where the most pressing issue in my opinion.
> >
> > I, furthermore, agree with the comments on the linearity assumptions in that paper. While the authors rightly argue that Eq. 5 only poses a particular possible implementation, it is the only one shown and tested in this paper. I would like to recommend that the authors include the provided comments regarding possible implications of their choice in the final version.
> >
> > Q2.3 / Variable Identifiability: What I meant in my initial review, was whether the presented method can identify and validate the correct working of intermediate mechanisms that might be, in turn, relevant for the final model output? While in their particular setting the authors are primarily concerned with the causal correctness of the final target variable, considerations on *how* models come to  their conclusions might further strengthen explanations on the identified mechanisms.
> >
> > The remaining points only pose minor concerns. I have, therefore, raised my score to an accept.

---

> > > ### Author Response · Authors · 2025-04-05
> > >
> > > We want to thank you again for your detailed feedback that has helped sharpen the presentation of our work!
> > >
> > > ---
> > > > I would like to recommend that the authors include the provided comments regarding possible implications of their choice in the final version.
> > >
> > > Thank you for the suggestion! We will clarify our assumptions and implications of our methods in the final version of the paper.
> > >
> > > > Q2.3 / Variable Identifiability: What I meant in my initial review, was whether the presented method can identify and validate the correct working of intermediate mechanisms that might be, in turn, relevant for the final model output? While in their particular setting the authors are primarily concerned with the causal correctness of the final target variable, considerations on how models come to their conclusions might further strengthen explanations on the identified mechanisms.
> > >
> > > This is a great point, and we share the reviewer’s enthusiasm about the possibility of gaining further clarity about how models are generating final outputs, especially in the “open-ended” tasks where we currently have only partial understanding.
> > >
> > > For instance, in MMLU we rely purely on specific multiple-choice mechanisms from Wiegreffe et al. (2025). Notably, this is already enough to show an improvement in correctness prediction. But a fuller understanding of the mechanism mediating between inputs and outputs could potentially lead to even greater improvements, in addition to other benefits.

---

### Decision · Program_Chairs · 2025-05-01

**Decision:**

Accept (poster)

**Comment:**

This work is nicely summarized by Reviewer w7xm and I quote, "The authors try to identify internal features of LLM that mediate causal effects on the final prediction output. In settings where LLM predictions align with the actual real-world causal process, causal features are assumed to resemble the ground-truth mechanism and, therefore, allow for predictions that are invariant to external disturbances or out-of-distribution queries." Two methods are proposed, which are then evaluated across a diverse suite of tasks under both in-distribution and OOD settings.

Overall, the reviewers were very happy with the submission and positive regarding the contributions. The paper nicely balances the diverse tasks within language modeling along with the complexity of presentation and clarity of writing. The rebuttal further clarified the issues of the reviewers and strengthened the overall story of the work. Even the most critical reviewer explicitly stated that they are not opposed to acceptance. I would still advise the authors to carefully take into account all the comments by the reviewers and make the necessary changes in the final version of the manuscript.

The AC recommends acceptance.